# Photoelectrochemical oxidative C(sp$^3$)−H borylation of unactivated hydrocarbons

Ping-Fu Zhong[1], Jia-Lin Tu[1], Yating Zhao[2], Nan Zhong[1], Chao Yang [1], Lin Guo[1] ✉ & Wujiong Xia [1,3] ✉

Organoboron compounds are of high significance in organic synthesis due to the unique versatility of boryl substituents to access further modifications. The high demand for the incorporation of boryl moieties into molecular structures has witnessed significant progress, particularly in the C(sp$^3$)−H borylation of hydrocarbons. Taking advantage of special characteristics of photo/electro-chemistry, we herein describe the development of an oxidative C(sp$^3$)−H borylation reaction under metal- and oxidant-free conditions, enabled by photoelectrochemical strategy. The reaction exhibits broad substrate scope (>57 examples), and includes the use of simple alkanes, halides, silanes, ketones, esters and nitriles as viable substrates. Notably, unconventional regioselectivity of C(sp$^3$)−H borylation is achieved, with the coupling site of C(sp$^3$)−H borylation selectively located in the distal methyl group. Our method is operationally simple and easily scalable, and offers a feasible approach for the one-step synthesis of high-value organoboron building blocks from simple hydrocarbons, which would provide ample opportunities for drug discovery.

Recent years have witnessed a fast development of visible light mediated photoredox catalysis[1-5], which can allow a series of thermo-dynamically inaccessible or kinetically inert reactions to occur. Although remarkable progress has been already made, stoichiometric terminal redox agents are always required and undesired byproducts are generated, both of which complicate the photochemical reactions. On the other hand, electrochemistry provides an intriguing approach to this issue by utilizing electricity as traceless redox agent[6-8]. The simultaneous occurrence of anodic oxidation and cathodic reduction enables a vast variety of mild radical-mediated transformations, obviating the need for stoichiometric electron or proton acceptors. Considering the advantages and common features of photoredox catalysis and electrochemistry, the merging of these two synthetic techniques has significantly enriched the catalytic strategies for the exploration of more challenging chemical transformations[9-12]. Pioneering work of photoelectrocatalysis by Xu[13-19], Lambert[20-28], Lin[29,30], Wickens[31], Lei[32,33], and others[34-42] has demonstrated the successful combination of photocatalysis and electrochemistry to unlock novel

reactivity in organic synthesis. From a mechanistic point of view, photoelectrocatalysis can be generally divided into two types of reaction process: anodic oxidation and cathodic reduction (Fig. 1a), and the two systems of photocatalysis and electrolysis remain completely independent and can be modified at will.

Over the past few decades, C(sp$^3$)−H functionalization of hydro-carbons has emerged as a vibrant research field in organic synthesis[43-48]. Selective functionalization of unreactive C(sp$^3$)−H bonds, which are widespread in abundant and inexpensive chemical feedstock, to produce high value-added chemicals is not only highly rewarding in organic chemistry but also yet to be fully explored. Considering the essential role of organoboron compounds as key building blocks in organic synthesis[49-53], the selective installation of boryl substitutions into the structure of hydrocarbons continuously attracts chemists' attention[54]. Not surprisingly, the recent emergence of elegant research work on C(sp$^3$)−H borylation reaction has provided new strategies for the access to organoborons[55-68]. For instance, the Hartwig group developed an undirected C(sp$^3$)−H borylation method

[1]State Key Lab of Urban Water Resource and Environment, Harbin Institute of Technology (Shenzhen), Shenzhen 518055, China. [2]College of Chemical and Material Engineering, Quzhou University, Quzhou 324000, China. [3]School of Chemistry and Chemical Engineering, Henan Normal University, Xinxiang, Henan 453007, China. ✉e-mail: guolin@hit.edu.cn; xiawj@hit.edu.cn

### a) Photoelectrochemical SET oxidation and reduction

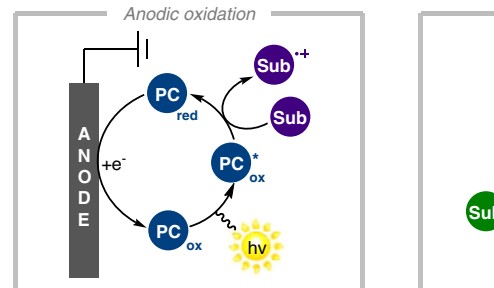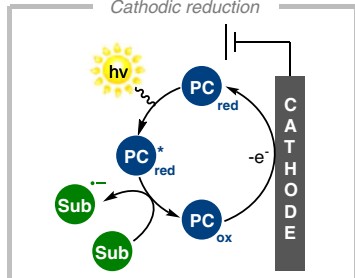

### b) Current strategies for C(sp³)-H borylation

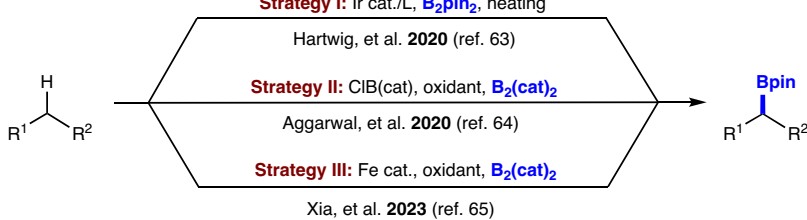

### c) Photoelectrochemical C(sp³)-H borylation (*this work*)

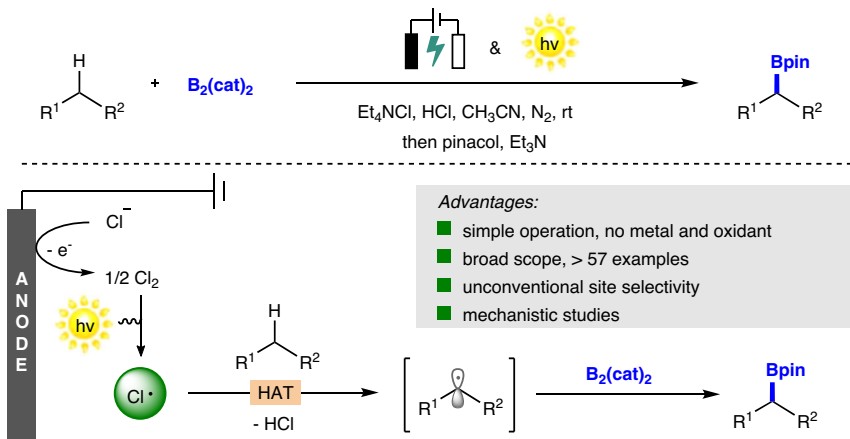

**Fig. 1 | Photoelectrochemical C(sp³)−H borylation. a** Photoelectrochemical SET oxidation and reduction. **b** Current strategies for C(sp³)-H borylation. **c** This work.

enabled by iridium catalyst and ligand[63]. Aggarwal and co-workers recently demonstrated a metal-free approach for selectively installing a boryl group on C(sp³)−H bond of hydrocarbons via photoinduced hydrogen atom transfer (HAT) approach[64]. Our group also reported an iron-catalyzed C(sp³)−H borylation protocol enabled by the photo-induced ligand-to-metal charge transfer (LMCT) process (Fig. 1b)[65].

Considering the significance of organoboron motifs in chemistry, we envision that the exploration of a general and sustainable platform for undirected C(sp³)−H borylation reaction via photoelectrocatalysis would be highly rewarding. Inspired by Xu's research work[14], the proposed photoelectrochemical method, which involves the generation of key chlorine radicals (Cl·) by anodic oxidation of Cl⁻ as well as the following homolytic photolysis, can potentially provide an ideal platform for C(sp³)−H borylation via radical-mediated HAT process (Fig. 1c)[69–76].

In this work, we report an oxidative photoelectrochemical strategy that allows the C(sp³)−H borylation reaction of hydrocarbons to smoothly occur. This synthetic protocol is applicable not only to simple alkanes, but also to a wide range of hydrocarbons including halides, silanes, ketones, esters, and nitriles. Importantly, unconventional regioselectivity of C(sp³)−H borylation is herein achieved, with the occurrence of C(sp³)−H borylation preferentially at distal methyl position.

## Results

### Optimization studies

Our study began by the investigation of the reaction between cyclo-hexane (**1**) and bis(catecholato)diboron (B₂(cat)₂, **2**), in a simple undivided cell equipped with a graphite rod as anode and a platinum plate as cathode (Table 1). After an extensive evaluation of the reaction conditions, the corresponding borylated product (**3**) was afforded in 72% yield (entry 1), employing tetraethylammonium chloride as electrolyte and chlorine source, HCl as the additive in acetonitrile under 390 nm purple LEDs irradiation as well as 2 mA constant current electrolysis for 12 h at room temperature. As shown in entries 2–4, unsatisfying results were obtained in the absence of HCl, and the use of FeCl₃ or CuCl₂ instead of HCl showed little reactivity. Moreover, replacing the platinum plate cathode with graphite rod cathode, or replacing the graphite rod anode with a platinum plate anode led to relatively lower efficiency (entries 5–6). The experimental results showed that decreasing or increasing the constant current is unfavorable to the reaction (entries 7-8). Furthermore, the use of other solvents, such as DMF, CF₃OH, CF₃CH₂OH, and DMSO, was explored but provided no improvement over CH₃CN (entries 9–12), and the reaction using *n*Bu₄NCl as electrolyte instead of Et₄NCl resulted in only trace of product **3** (entry 13). In addition, the yield of alkyl boronate **3** slightly decreased when the reaction was performed using 5.0 equiv. of

## Table 1 | Optimization Studies [a]

| Entry | Variations from standard conditions | 3 (%)[b] |
|---|---|---|
| 1 | None | 72 (68[c]) |
| 2 | Without HCl | trace |
| 3 | FeCl$_3$ (20 mmol%) instead of HCl | trace |
| 4 | CuCl$_2$ (20 mmol%) instead of HCl | trace |
| 5 | C(+)/C(−) instead of C(+)/Pt(−) | 61 |
| 6 | Pt(+)/Pt(−) instead of C(+)/Pt(−) | 46 |
| 7 | I = 1.5 mA | 32 |
| 8 | I = 2.5 mA | 26 |
| 9 | DMF instead of CH$_3$CN | 14 |
| 10 | CF$_3$OH instead of CH$_3$CN | 39 |
| 11 | CF$_3$CH$_2$OH instead of CH$_3$CN | 27 |
| 12 | DMSO instead of CH$_3$CN | 10 |
| 13 | nBu$_4$NCl instead of Et$_4$NCl | trace |
| 14 | 5.0 equiv. of cyclohexane 1 | 58 |
| 15 | TFA/TfOH instead of HCl | N.D. |
| 16 | Without electricity | N.D. |
| 17 | Without light irradiation | N.D. |

[a]Standard conditions: undivided cell, graphite rod anode (φ = 3 mm), Pt plate cathode (10 × 10 × 1 mm), cyclohexane 1 (2.0 mmol, 10.0 equiv.), B$_2$(cat)$_2$ 2 (0.2 mmol, 1.0 equiv.), Et$_4$NCl (0.1 mmol, 0.017 M), concentrated HCl (0.8 mmol, 4.0 equiv.), CH$_3$CN (6.0 mL, technical grade), 390 nm LEDs (10 W), constant current = 2 mA under N$_2$ atmosphere at room temperature for 12 h (4.48 F/mol); then pinacol (0.6 mmol, 3.0 equiv.) and triethylamine (0.84 mL) in DCM (1.0 mL), 1 h. [b]Yields determined by NMR analysis using 1,3,5-(OMe)$_3$C$_6$H$_3$ as the internal standard. [c]Yields of isolated products. N.D. Not detected.

cyclohexane 1 (entry 14). When using trifluoroacetic acid (TFA) or trifluoromethanesulfonic acid (TfOH) instead of HCl for the reaction, the target product cannot be obtained at all (entry 15). As anticipated, control experiments revealed that electric current and light irradiation were both critical for the desired transformation (entries 16-17).

## Exploration of substrate scope

Under the existing optimization conditions, we began to study the reaction of various C(sp$^3$)−H species with B$_2$(cat)$_2$ under photoelectrochemical conditions to explore the range of viable substrates. As shown in Fig. 2, in addition to cyclohexane, other cycloalkanes, including cyclopentane (4), cycloheptane (5), cyclooctane (6) and cyclododecane (7), are all found to be suitable C(sp$^3$)−H substrates, which were smoothly converted into the corresponding boronate ester products in 50−67% yields. It is noteworthy that adamantane (8) reacts with B$_2$(cat)$_2$ in a unconventionally regioselective manner. The rigid caged structure of adamantane have led to a series of C−H functionalization protocols to proceed at the slightly stronger 3 °C−H bond (BDE of 3 °C−H bond: 99 kcal/mol; BDE of 2 °C−H bond: 96 kcal/mol)[77,78]. In contrast, our photoelectrochemical C(sp$^3$)−H borylation reaction preferentially delivered the 2 °C−H borylated product 8 in moderate yield (47%) and 86:14 regioselectivity (2°:3°). Several acyclic alkanes were tested and were found to be well compatible under standard conditions, furnishing the desired alkyl boronate esters 9−12 in moderate yields. Interestingly, the site selectivity matched the Aggarwal's results[64], exclusively occurring at the distal methyl position rather than weaker 3 °C(sp$^3$)−H bonds, which indicates that steric hindrance plays a key role in affecting the regioselectivity of C−H borylation. Furthermore, borylation of n-pentane gave 13 in a

combined yield of 63%. The reaction favored the distal methyl group over the methylene groups, and resulted in 54:33:13 selectivity (α:β:γ), which is also similar to Aggarwal's results. Another interesting example is norbornane, which could react with B$_2$(cat)$_2$ to give borylated product 14 in highly regioselective and stereoselective manner. Chlorine-containing molecules were found to be viable substrates (15−17), with the observation of minor dehalogenation product (usually <10%). In addition, isopropyl-substituted benzene were borylated with complete site selectivity for the methyl groups over both aromatic and benzylic C−H bonds (18−19).

Next, we studied the range of oxidative photoelectrochemical C(sp$^3$)−H borylation of silicon-containing molecules. C(sp$^3$)−H borylation was successful at the α-silyl position of trimethyl(phenyl)silane (20), benzyltrimethylsilane (21), 1,4-bis(trimethylsilyl)benzene (22), 1,1,1,2,2,2-hexamethyldisilane (23), and bis(trimethylsilyl)methane (24), delivering synthetic valuable (borylmethyl)silanes in 44−69% yields. The same trend was also observed for the utilization of halogen-containing silanes (25−27). Notably, the C(sp$^3$)−H borylation of tetraethylsilane (28) revealed that steric effect still dominates the site selectivity, preferentially giving the distal methyl borylated products over the α-borylated ones (α:β = 12:88). The C−H borylation of boryl group-containing compound gave 30 in 46% yield with complete site selectivity for the less-hindered primary C−H sites. Moreover, selective C(sp$^3$)−H borylation of ether compounds, such as 1-(isopentyloxy)−3-methylbutane (31) and 1-propoxypropane (32), was also obtained, albeit in a decreased yield. Unlike conventional C−H functionalization protocols that exhibited high selectivity at the α-position to oxygen[79,80], our method allowed C−H borylation to occur preferentially at the primary methyl position, with no borylation of α-

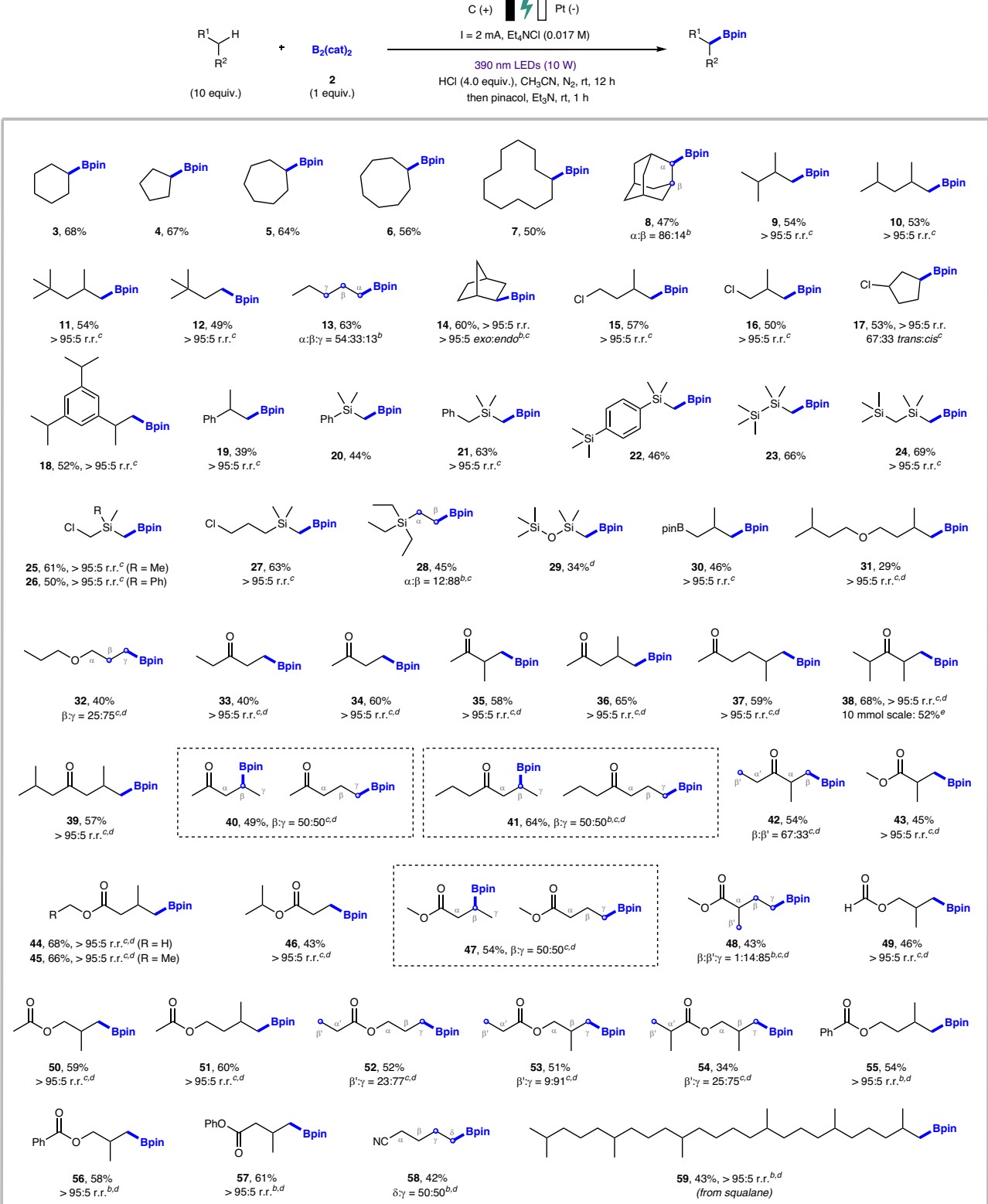

**Fig. 2 | Substrate scope.** Reaction conditions: undivided cell, graphite rod anode (φ = 3 mm), Pt plate cathode (10 × 10 × 1 mm), substrate (2.0 mmol, 10.0 equiv.), B₂(cat)₂ 2 (0.2 mmol, 1.0 equiv.), Et₄NCl (0.1 mmol, 0.017 M), concentrated HCl (0.8 mmol, 4.0 equiv.), CH₃CN (6.0 mL, technical grade), 390 nm LEDs (10 W), constant current = 2 mA under N₂ atmosphere at room temperature for 12 h (4.48 F/mol); then pinacol (0.6 mmol, 3.0 equiv.) and triethylamine (0.84 mL) in DCM (1.0 mL), 1 h. [a] Yields of isolated products. [b] Regioselectivity (r.r.) and diastereomeric ratio (d.r.) were determined by ¹H NMR analysis. [c] r.r. and d.r. were determined by GC analysis. [d] Using HCl in 1,4-dioxane (4.0 M) instead of concentrated HCl, constant current = 1.7 mA, and using anhydrous CH₃CN instead of technical grade CH₃CN. [e] Large-scale reaction: undivided cell, graphite rod anode (φ = 6 mm), Pt plate cathode (10 × 10 × 1 mm), 390 nm LEDs (30 W), constant current = 5 mA for 24 h.

C(sp³)–H bond observed. This is probably because the C(sp³)–H bonds adjacent to an oxygen atom can be further oxidized to form the corresponding carbocations under electrochemical conditions[81,82].

The ubiquity of ketones has demonstrated the significance of these building blocks in organic chemistry. However, the introduction of a boryl group into ketone motifs has been rarely reported. Different types of carbonyl compounds were selected to test the adaptability of our photoelectrochemical C(sp³)–H borylation method. As expected, the reaction of several simple ketones as substrates gave the corresponding borylated products **33–39** in moderate to good yields and a complete site selectivity for the less-hindered distal methyl sites. The regioselectivity seems to be very sensitive to the carbonyl group, with no borylation observed at its α-position in all the cases. When the use of pentan-2-one (**40**) and heptan-4-one (**41**) was investigated under the standard conditions, the corresponding products can be obtained without obvious preference (β:γ = 50:50), probably due to an effect of boron–carbonyl-oxygen interaction[83,84]. Likewise, the C(sp³)–H borylation of 2-methylpentan-3-one (**42**) gave no preference to the β- and β′-positions. Moreover, esters also proved to be viable substrates, and again C–H borylation preferentially occurred at distal methyl position (**43–46**). Although the site selectivity of substrates such as methyl butyrate (**47**) showed the same trend as pentan-2-one, steric hindrance still have significant effect on the site selectivity, allowing methyl 2-methylbutanoate (**48**) to proceed with the γ-C–H borylated product being the most favored. When the isopropyl end was far away from the aliphatic ester functional group, the reaction showed good yields as well as regioselectivity (**49–51**). In addition, propyl propionate (**52**), isobutyl propionate (**53**), and isobutyl isobutyrate (**54**) were selected as suitable substrates to further investigate the selectivity of ester-containing compounds. Our experimental results revealed the same trend of regioselectivity, affording C(sp³)–H borylation products **52–54** with coupling site preferentially at the distal methyl group on the ester oxygen side. Similar results were observed for the use of phenyl-containing esters as substrates, with the exclusive installation of boryl moiety at the terminal isopropyl group (**55–57**). In addition, the extension of our photoelectrochemical protocol to nitriles was also successful in affording cyano-containing organoboron product **58** in 42% yield and 50:50 selectivity (δ:γ). The limitation of this photoelectrochemical protocol is that the reaction cannot be well applied to

nitrogen-containing substrates, such as amides, amines, or sulfonamides. Late-stage modification of complex organic molecules is the basis for the evaluation of a practical protocol. The direct use of B₂(cat)₂ with complex hydrocarbons to alkylboron units makes the standard photoelectrochemical conditions applicable for the late-stage modification of biologically valuable natural products and pharmaceuticals. For example, squalane, a saturated oil that has anti-inflammatory property and is often utilized in skin care products as a moisturizer, was successfully borylated at the terminal methyl site to give alkyl boronate product (**59**) in 43% yield and complete site selectivity.

## Gram-scale reaction

In order to further demonstrate the synthetic potential of this photoelectrochemical method, a gram-scale experiment was performed on the batch. When photo-irradiation and electrolysis were carried out with modified light source, electrode materials, and reaction conditions using 10 mmol B₂(cat)₂ and 100 mmol 2,4-dimethylpentan-3-one as substrates, 1.25 g (52% yield) of the desired β-borylated product **38** was successfully obtained (Fig. 2), which indicates that it represents a promising synthetic method and has the potential to easily transform two readily available starting materials into highly valuable organoboron compounds. Subsequently, we also successfully translated our batch protocol to a scalable continuous-flow fashion, which enables a larger scale of transition between cyclohexane (**1**) and B₂(cat)₂ (**2**) under modified photoelectrochemical conditions (Fig. 3). Although the reaction time may not be greatly shortened (very slow flow rate was applied due to the restriction of our continuous-flow setup), the collected reaction solution smoothly afforded the borylated product (**3**) in 61% yield on a 20 mmol scale, after workup and purification. By applying a flow rate of 0.1 mL/min in flow, we were also able to prepare alkylboron product **38** in 56% yield and complete distal methyl selectivity.

## Mechanistic studies

Cyclic voltammetry (CV) was then performed to add more credence to the above results. The battery device was composed of glassy carbon working electrode, Pt coil pair electrode and Hg/HgCl (in saturated KCl solution) reference electrode. All measurements were carried out in

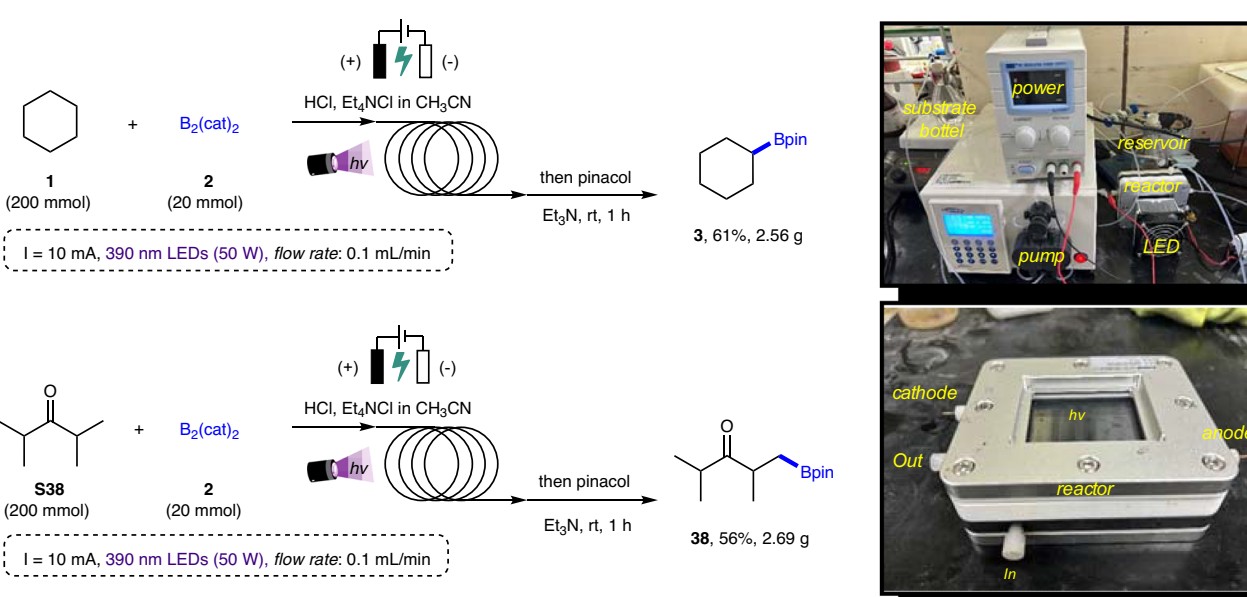

**Fig. 3 | Gram-scale reaction in continuous-flow.** Reaction conditions: substrate **1** or **S38** (200 mmol, 10 equiv.), B₂(cat)₂ **2** (20 mmol, 1.0 equiv.), Et₄NCl (10 mmol), concentrated HCl (60 mmol), CH₃CN (150 mL, technical grade), 390 nm LEDs (50 W), constant current = 10 mA under N₂ atmosphere at room temperature (flow rate: 0.1 mL min⁻¹); then pinacol (60 mmol, 3.0 equiv.) and triethylamine (400 mmol, 20 equiv.) in DCM (20 mL), 1 h.

dry degassed MeCN, using 0.1 M $n$Bu$_4$NClO$_4$ as the supporting electrolyte, and each measurement consisted of an oxidation scan followed by a reverse reduction scan using a 100 mV/s scan rate. As shown in Fig. 4, it can be seen from the CV diagram that the oxidation potential of HCl is 1.87 V $vs.$ SCE, the oxidation potential of chlorine is 1.62 V, and the oxidation potential of B$_2$(cat)$_2$ (2) is 2.20 V. From these results we can realize that Cl$^-$ species is more susceptible to oxidation when compared with B$_2$(cat)$_2$, suggesting that the anodic oxidation process is more likely to initiate with the oxidation of Cl$^-$ to generate chlorine radical intermediate, rather than the oxidation of B$_2$(cat)$_2$ (2).

To gain more insights into the reaction mechanism, a series of mechanistic experiments were conducted, as shown in Fig. 5. According to Xu's work[14], the C(sp$^3$)–H borylation of cyclohexane (1) with B$_2$(cat)$_2$ (2) in the presence of Cl$_2$, which was generated by mixing NaOCl with HCl, provided the desired alkyl boronate ester 3 in 28% yield under 390 nm light irradiation. However, the formation of 3 was not observed in the absence of LED light (Fig. 5a), which revealed the involvement of Cl$_2$ formation in our photoelectrochemical C(sp$^3$)–H borylation reaction, and the Cl$_2$ is produced through electrochemical process. To explore whether the HAT process is the rate-determining step of the whole photoelectrochemical process, a kinetic isotope effect (KIE) experiment was subsequently carried out, as shown in Fig. 5b. The reaction of cyclohexane (1) and deuterated cyclohexane (1-$d_{12}$) with B$_2$(cat)$_2$ delivered a mixture of products 3 and isotopically labeled 3-$d_{11}$ in 64% yield with secondary kinetic isotope effects ($k_H/k_D = 1.0$), indicating that the HAT process is not a turnover limiting step for this reaction. Furthermore, the electrolysis of styrene (60) in the absence of boron reagent afforded a chlorine-containing ketone product 61 in 69% yield, indicating the generation and involvement of chlorine radicals during our reaction process (Fig. 5c). Consistent with this finding, the reaction of $N,N$-diallyltosylamine (62) under the photoelectrochemical system produced the chlorinated ring-closing product 63, which is derived from a chlorine radical addition–5-$exo$-trig cyclization–hydrogenation sequence[85]. In order to capture the key radical intermediates, an electron paramagnetic resonance (EPR) experiment was performed by adding a free radical spin trap 5,5-dimethyl-1-pyrroline $N$-oxide (DMPO). The afforded EPR spectroscopy revealed that carbon-centered radicals are generated via intermolecular HAT process and rapidly captured by DMPO to form relatively stable free radicals ($g = 2.007$, $A_N = 14.81$ G, and $A_{H\beta} = 21.38$ G)[86–89].

After demonstrating the key intermediates during the reaction process, we next started to investigate the origin of the generated site selectivity for the C(sp$^3$)–H borylation. In general, C(sp$^3$)–H functionalization reactions via chlorine radical-mediated HAT process preferentially occur at the weaker C–H bonds, leading to the regioselectivities that follow the trend of 3° > 2° > 1°[87,90–92]. Considering that the chlorine radical-mediated HAT step has proved to be not turnover-limiting through our KIE experiment, we hypothesized that chlorine radical performs the HAT process to initially give a 3° carbon-centered radical. However, slower reactivity of more substituted radical with B$_2$(cat)$_2$ may result in faster functionalization of the sterically unhindered 1° radical, which implies that the alkyl radical formation via HAT is fast and reversible. To test our hypothesis, three deuterated compounds (S55-$d$, S56-$d$, and S57-$d$) were synthesized, with moderate to high levels of deuterium incorporation at the tertiary sites (Fig. 5e)[93]. As expected, upon light irradiation and electrolysis for 12 h moderate H/D scrambling was observed, with decreased D-incorporation at the tertiary sites of the corresponding C–H borylated products (55-$d$, 56-$d$, and 57-$d$). Furthermore, 2,3-dimethylbutane (DMB, S9) was utilized as a standard alkane substrate to probe the site selectivity (Fig. 5f)[87,94,95]. Under the standard photoelectrochemical conditions, control reaction revealed that the addition of benzylidenemalononitrile (64)[96] as a typical radical acceptor greatly inhibited the desired C–H borylation reaction, while generating C–H alkylated product 65 preferentially at the weaker 3 °C–H bond (intrinsic selectivity of 1°/3°: 1:3). Consistent with this finding, the addition of phenyl acrylate (66)[79] into the photoelectrochemical system with DMB (S9) and B$_2$(cat)$_2$ (2) produced C–H alkylated product 67 in 65% yield, and again, the alkylation at the 3 °C–H bond is preferred (intrinsic selectivity of 1°/3°: 1:6). These results supported the reversible HAT process to form carbon-centered radicals, and implied that site selectivity is largely affected by the use of radical acceptors[65].

In order to realize the HAT species under our photoelectrochemical conditions, more mechanistic experiments were further carried out (see the Supplementary Information for details). The experimental results indicate that free Cl radical and Cl-boron complexes are both likely to be the HAT species during the reaction process[97]. To further understand the role of HCl in this photoelectrochemical system, several control experiments were carried out, as shown in Fig. 5g. Replacing HCl with other inorganic chloride salts, such as NaCl, CsCl, and LiCl, was found to be ineffective. Although the

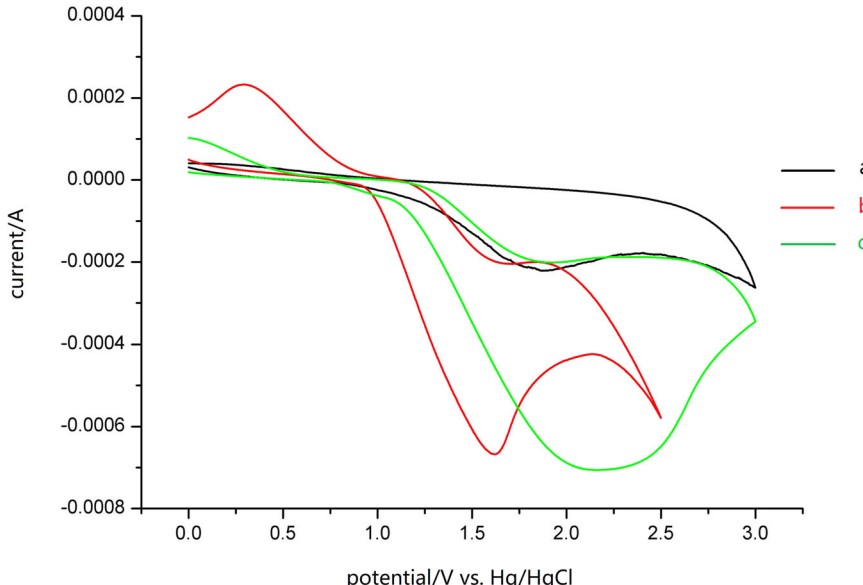

**Fig. 4 | Cyclic voltammograms.** The cyclic voltammogram was measured in MeCN ($n$Bu$_4$NClO$_4$). a HCl (2 mmol L$^{-1}$); b Et$_4$NCl (2 mmol L$^{-1}$); c B$_2$(cat)$_2$ 2 (10 mmol L$^{-1}$).

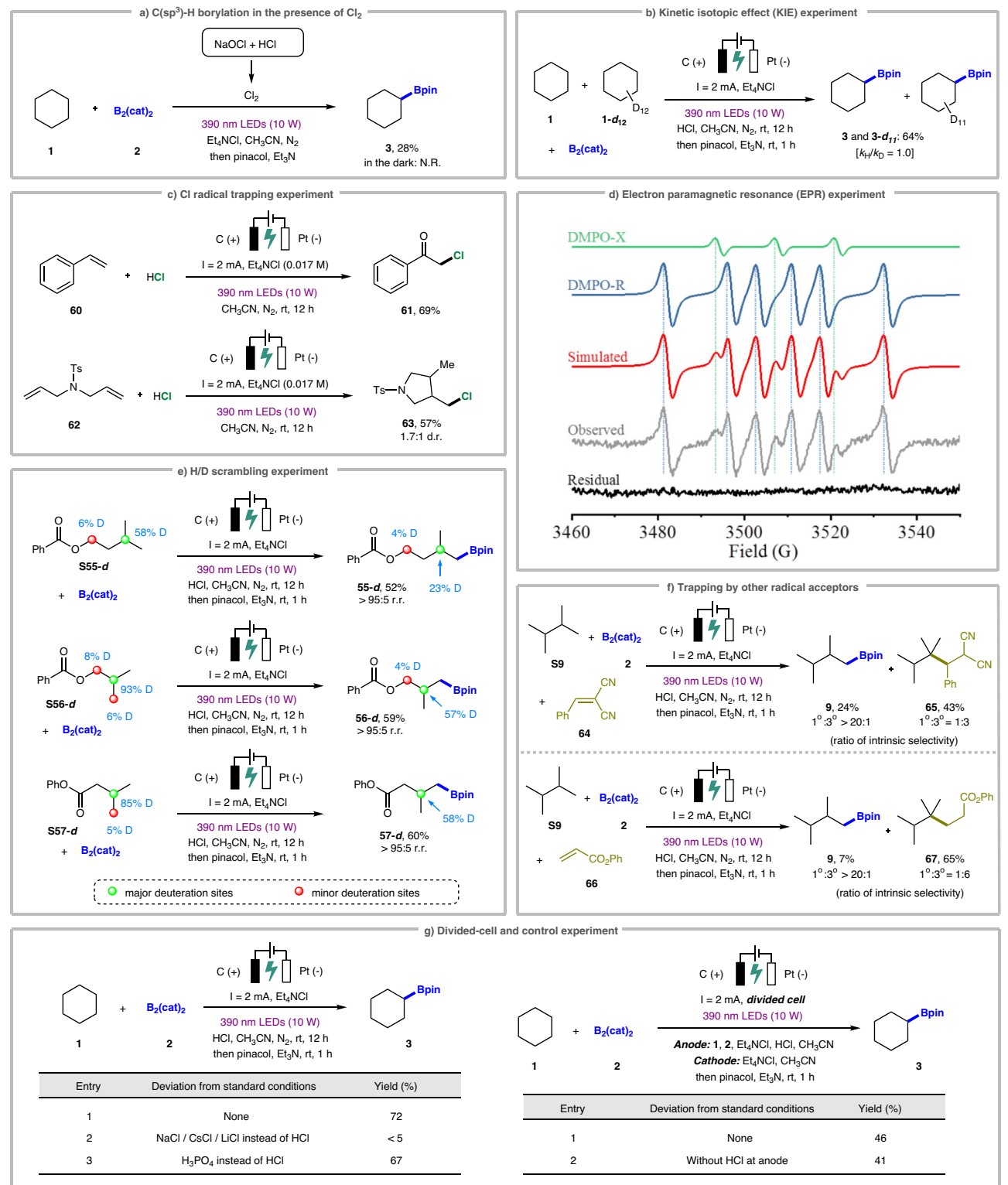

**Fig. 5 | Mechanistic studies. a** C(sp³)-H borylation in the presence of Cl₂, **b** Kinetic isotopic effect (KIE) experiment, **c** Cl radical trapping experiment, **d** EPR experiment, **e** H/D scrambling experiment, **f** Trapping by other radical acceptors, **g** Divided-cell and control experiment.

use of other acids instead of HCl, such as TFA and TfOH, led to complete loss of reaction efficiency (Table 1, entry 15), substituting HCl by H₃PO₄ successfully gave the desired C−H borylated product **3** in 67% yield, which indicates that HCl mainly plays a crucial role as proton source rather than chlorine source[14]. At the cathode surface, protons undergo cathodic reduction to generate H₂, and protect the substrate B₂(cat)₂ from single-electron reduction in certain conditions[98].

Moreover, a divided-cell experiment was carried out (Fig. 5g). Substrates **1** and **2**, along with Et₄NCl, HCl, and CH₃CN, was placed into the anode chamber, while Et₄NCl and CH₃CN were added into the cathode chamber. As expected, upon light irradiation and electrolysis for 12 h the desired product **3** was detected in the anode chamber with 46% yield, suggesting that the photoelectrochemical C(sp³)−H borylation occurred surrounding the graphite rod anode. Interestingly, a

moderate yield (41%) of alkyl boronate **3** was still afforded in the absence of HCl at the anode chamber, probably because the $B_2(cat)_2$ reagent in the anode chamber of divided-cell would not be electrochemically reduced, and therefore it does not need the protection of HCl any more. The protons that come from trace amount of $H_2O$ in the cathode cell are combined with electrons at the cathode to generate $H_2$.

Based on the above mechanistic studies and relevant literature reports, we propose the plausible mechanism outlined in Fig. 6. First, the $Cl^-$ is electrochemically oxidized to $Cl_2$ at the anode surface, and then the chlorine radical species (**A**) is generated via light-promoted homolytic cleavage of $Cl_2$. According to its inherent selectivity, the chlorine radical undergoes HAT process with $C(sp^3)$–H compounds (taking **S9** as an example) to initially release a more substituted carbon-centered radical (**B**), which could not proceed with the C–H borylation reaction at the tertiary site, probably due to steric hindrance. Since the alkyl radical formation is reversible (or the alkyl radical can "isomerize" by performing the subsequent reversible HAT with other molecules of the substrate) and fast, a sterically unhindered primary radical (**D**) is generated and trapped by $B_2(cat)_2$ to give an alkyl boronate ester (**E**) as well as the ligated boryl radical (**F**) (*Path A*). Alternatively, the HO-Bcat (**G**) generated by hydrolysis and anodic oxidation in the reaction system can be complexed with Cl radical species (**A**) to obtain Cl-radical-boron "ate" complex (**H**), followed by the HAT process to obtain carbon center radical (**D**) (*Path B*).

Treating intermediate **E** with pinacol and triethylamine finally delivers the desired product **9**. At the cathode surface, protons undergo cathodic reduction to generate $H_2$, obviating the need for sacrificial oxidants.

## Discussion

In summary, we have developed an efficient method that is capable of preparing different types of aliphatic boronate ester products from readily available hydrocarbons and $B_2(cat)_2$ via photoelectrochemical $C(sp^3)$–H borylation reaction. Without the need of metal catalysts or oxidants, our protocol addresses the limitations of narrow substrate scope for previously reported $C(sp^3)$–H borylation reactions. The scope of our method has been remarkably expanded (>57 examples), and includes the use of simple alkanes, halides, silanes, ketones, esters and nitriles as viable substrates. Importantly, the reaction exhibits unconventional site selectivity, with the occurrence of $C(sp^3)$–H borylation preferentially at distal methyl position. In addition, a gram-scale synthesis of alkylboron product has been performed on a 20 mmol scale in continuous-flow, which greatly highlights the synthetic potential of our method. Detailed mechanistic studies and discussion of the unconventional site selectivity has been also carried out. Given the significance and versatility of organoboron compounds as valuable building blocks in organic synthesis as well as unconventional site selectivity we are reporting, we believe that this study will offer enormous opportunities for selectively installing valuable functional groups of drug molecules at the late stage.

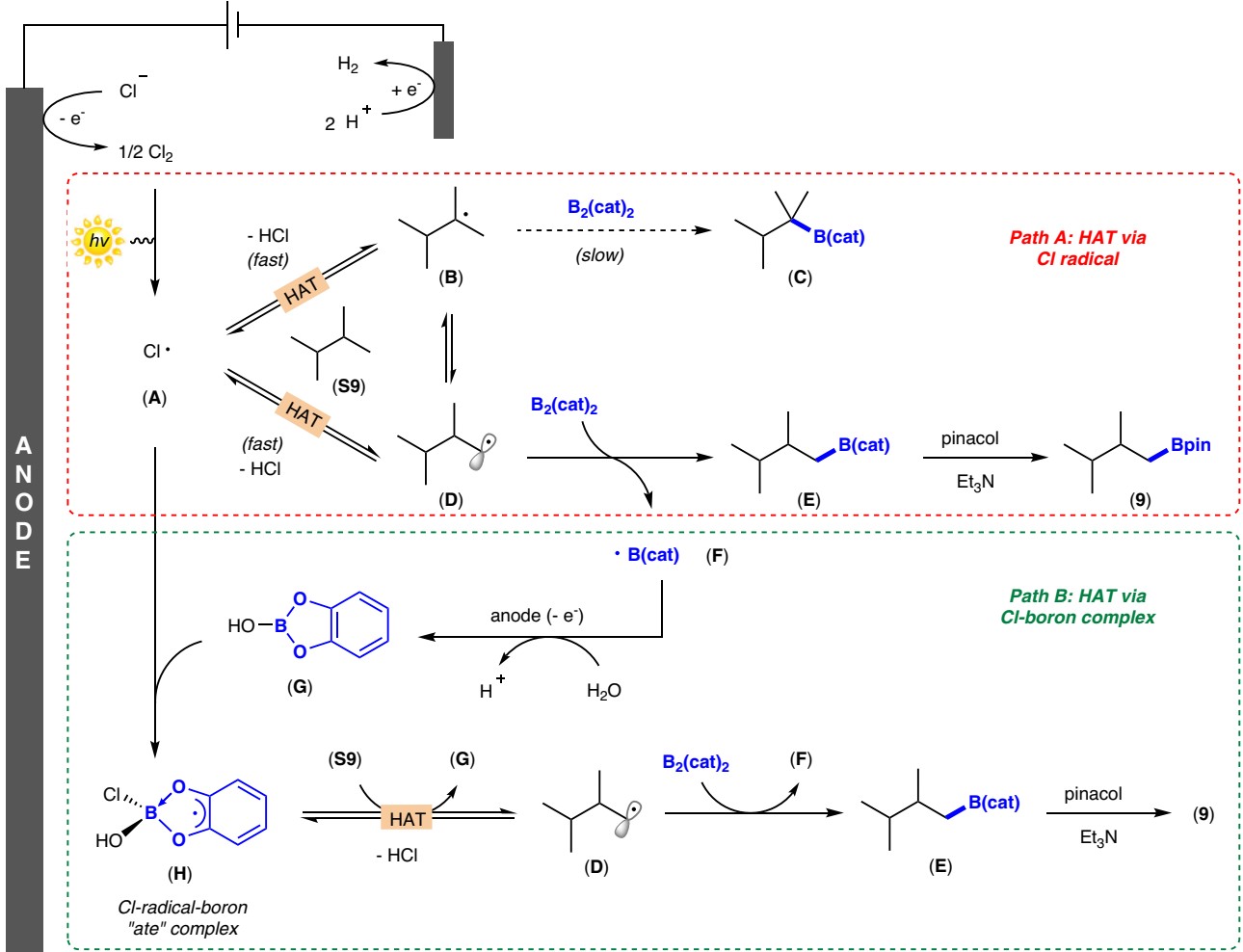

**Fig. 6 | Plausible reaction mechanism. Path A** HAT via Cl radical, **Path B** HAT via Cl-boron complex.

## Methods

### General procedure A for photoelectrochemical C(sp$^3$)−H borylation (alkanes, halides, and silanes)

To a 10 mL vial equipped with a stir bar was added B$_2$(cat)$_2$ (48 mg, 0.20 mmol, 1.0 equiv.), Et$_4$NCl (0.1 mmol, 0.5 equiv.), and HCl (concentrated, 0.8 mmol, 4.0 equiv.). MeCN (6 mL) was then added followed by the alkane (2.0 mmol, 10 equiv.). The reaction mixture was stirred at 680 rpm, irradiated with 390 nm LED lamps and 2.0 mA electrolysis for 12 h. The reaction temperature was maintained at approximately room temperature by cooling with a desk fan. After irradiation, a solution of pinacol (71 mg, 0.60 mmol, 3.0 equiv.) and Et$_3$N (0.84 mL, 6.0 mmol, 30 equiv.) in CH$_2$Cl$_2$ (1 mL) was added and stirring was continued for 1 h. The reaction mixture was concentrated in vacuo and purified by flash column chromatography to afford the corresponding boronate ester product.

### General procedure B for photoelectrochemical C(sp$^3$)−H borylation (ethers, esters, and nitriles)

To a 10 mL vial equipped with a stir bar was added B$_2$(cat)$_2$ (48 mg, 0.20 mmol, 1.0 equiv.), Et$_4$NCl (0.1 mmol, 0.5 equiv.), and HCl in 1,4-dioxane (0.8 mmol, 4.0 equiv.). Dry MeCN (6 mL) was then added followed by the alkane (2.0 mmol, 10 equiv.). The reaction mixture was stirred at 680 rpm, irradiated with 390 nm LED lamps and 1.7 mA electrolysis for 12 h. The reaction temperature was maintained at approximately room temperature by cooling with a desk fan. After irradiation, a solution of pinacol (71 mg, 0.60 mmol, 3.0 equiv.) and Et$_3$N (0.84 mL, 6.0 mmol, 30 equiv.) in CH$_2$Cl$_2$ (1 mL) was added and stirring was continued for 1 h. The reaction mixture was concentrated in vacuo and purified by flash column chromatography to afford the corresponding boronate ester product.

## Data availability

Materials and methods, optimization studies, experimental procedures, mechanistic studies, $^1$H NMR spectra, $^{13}$C NMR spectra and mass spectrometry data generated in this study are provided in the Supplementary Information file. All other data are available from the corresponding author upon request.

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

## Acknowledgements

We are grateful for the financial support from the National Natural Sci-ence Foundation of China (No.22101066, L.G.), the Science and Tech-nology Plan of Shenzhen (No. JCYJ20210324133001004, W.X.; JCYJ20220531095016036, L.G.; and GXWD20220817131550002, W.X.), the Natural Science Foundation of Guangdong (No. 2022A1515010863, L.G.), Guangdong Basic and Applied Basic Research Foundation (No. 2021A1515220069, W.X.), and the Talent Recruitment Project of Guan-dong (No. 2019QN01L753, W.X.). The project was also supported by "the Fundamental Research Funds for the Central Universities" (Grant No. HIT.OCEF.2022039, L.G.), State Key Laboratory of Urban Water Resource and Environment (Harbin Institute of Technology) (No.2022TS23, W.X.), and the Open Research Fund of the School of Chemistry and Chemical Engineering, Henan Normal University (W.X.).

## Author contributions

P.-F.Z., L.G. and W.X. conceived and designed the project. P.-F.Z. per-formed and analyzed the experiments. J.-L.T., Y.Z., N.Z. and C.Y. con-tributed to the data analysis. L.G. wrote the manuscript. L.G. and W.X. supervised and directed the project.

## Competing interests

The authors declare no competing interests.
