## [Peer Review File · Nature Communications]

REVIEWER COMMENTS

Reviewer #1 (Remarks to the Author):

Professors Xia and Guo, and coauthors reported the development a photoelectro protocol for the oxidative C-H borylation. This method processed with a range of hydrocarbons with various functional groups, however, without examples of complex substrates. The proposed chlorine radical generation by the photoelectrochemical method is reported by Xu (ref 14), and the authors here applied for the subsequent HAT and carbon radical borylations. Probably, one of the most significant highlights of this research is the observed unconventional borylation regioselectivity of C(sp³)-H bonds, e.g. primary over tertiary borylation of several substrates in this research. Indeed, there is great need for the development of selective methodology for direct and selective borylation of C(sp³)-H bonds for the facile synthesis of organoboron compounds, especially for the ones processing unconventional selectivities. This manuscript presents a novel photoelectrochemical strategy for direct borylation of C(sp³)-H bonds and a relative broad scope is achieved with unconventional selectivities, however, this research is not considered as a completed investigation by the reviewer due to the lack of mechanistic studies and in-depth understanding of the unconventional selectivities. The reviewer recommends rejection of this version and suggests the authors to resubmit this manuscript if the following issues could be addressed.

The major concern of this research is raised from the discussion/interpretation of the unconventional selectivity and related reaction mechanism. The authors observed similar selectivity for several substrates as Aggarwal's published work (ref 64). And the authors rationalized such unconventional selectivity by steric effect only, e.g. primary over tertiary borylation. Accordingly, the authors proposed the generation of "free" chlorine radical for the following HAT. However, clearly the observed regioselectivities in the borylation indicate that "free" chlorine radicals are unlikely to be the predominant HAT species. (*J. Am. Chem. Soc.* 80, 4997–5001 (1958). *J. Org. Chem.* 63, 63, 8860–8864 (1998).)

Is it possible the radical chain mechanism is propagated by the reaction of chloride-stabilised boryl radical as the same or similar intermediate proposed by Aggarwal (ref 64). Obviously, more mechanistic studies should be conducted. For example, the authors could refer to Aggarwal's published work (ref 64).

As for the SI, more GC spectra should be provided to clarify the unconventional selectivity. More than 20 substrates showed the unconventional selectivity, however, only 8 examples are provided with GC spectra.

Reviewer #2 (Remarks to the Author):

The authors present an interesting approach to the oxidative C(sp³)-H borylation of hydrocarbons using photoelectrochemistry, eliminating the need for metals and oxidants. Organoboron compounds hold immense potential in organic synthesis, and the development of an efficient and versatile method for their synthesis is of high importance. One of the primary strengths of this study is the broad substrate scope and interesting regioselectivity of C(sp³)-H borylation, with a preference for the distal methyl position. The scalability of the procedure makes it even more appealing, as it facilitates access to high-value organoboron building blocks from simple hydrocarbons in a single step. I recommend accepting this article for publication after minor revisions. The authors should address the following points:

1. Provide more details on the mechanism of radical borylation and how it leads to the observed regioselectivity. The HAT process with chlorine atom is expected to be much less selective. Likely, only selected carbon radicals made their way to the organoboron products.
2. Add results with less hydrocarbons to Table 1. The use of large excess of hydrocarbons is a major limitation although simple alkanes can be recovered.
3. Explain the drastically different results with nBu₄NCl and Et₄NCl.
4. CF₃OH or CF₃CH₂OH?
5. CB(cat) was important for Aggarwal's work. It is important here?
6. There is no evidence that 57 is formed through addition of a chlorine atom to styrene.
7. What are the limitations?
8. 390 nm LED is purple LED not blue LED.
9. It is inaccurate to describe tetraethylammonium chloride as electrolyte and HCl as the chlorine source. They both provide chloride and increase conductivity. HCl is needed likely to increase proton reduction to avoid reduction of the boron reagent. What if you use other acids, such as TFA, TfOH.
10. Please give more details on the components of the photoelectrochemical cell to allow others to reproduce the results.

Reviewer #3 (Remarks to the Author):

The Manuscript by Xia, Guo and co-authors describes a terminal oxidant and transition metal-free, photoelectrochemical C(sp³)-H borylation. This method involves intermolecular hydrogen atom transfer by the chlorine radical, generated via anodic oxidation of chloride, and followed by photoinduced homolysis of the produced chlorine gas. The resulting alkyl radical was trapped with diboron reagent to afford alkyl boronates. Expectedly, this method exhibits high selectivity towards terminal C-H sites. This operationally simple and scalable protocol can be employed for broad range of substrates including halides, silanes, ketones and esters.

I think this work does not represent significant advances to meet the high standard of Nature Communications for the following reasons. This method involves known photoelectrochemical strategy (Ref: 14) for radical generation. It exhibits alike site-selectivity, as in previous radical C-H borylation reports (Ref: 64), and in the authors' recent work, which is not cited in this manuscript (J. Am. Chem. Soc. 2023, 145, 7600–7611).

REVIEWER COMMENTS AND RESPONSE

Reviewer #1

Professors Xia and Guo, and coauthors reported the development a photoelectro protocol for the oxidative C-H borylation. This method processed with a range of hydrocarbons with various functional groups, however, without examples of complex substrates. The proposed chlorine radical generation by the photoelectrochemical method is reported by Xu (ref 14), and the authors here applied for the subsequent HAT and carbon radical borylations. Probably, one of the most significant highlights of this research is the observed unconventional borylation regioselectivity of C(sp³)-H bonds, e.g. primary over tertiary borylation of several substrates in this research. Indeed, there is great need for the development of selective methodology for direct and selective borylation of C(sp³)-H bonds for the facile synthesis of organoboron compounds, especially for the ones processing unconventional selectivities. This manuscript presents a novel photoelectrochemical strategy for direct borylation of C(sp³)-H bonds and a relative broad scope is achieved with unconventional selectivities, however, this research is not considered as a completed investigation by the reviewer due to the lack of mechanistic studies and in-depth understanding of the unconventional selectivities. The reviewer recommends rejection of this version and suggests the authors to resubmit this manuscript if the following issues could be addressed.

Response: *We are very grateful for the reviewer's comments and useful advice. We have added very detailed mechanistic studies and in-depth understanding of the afforded site selectivity into the revised manuscript (please see Fig. 5 in the revised manuscript). For the lack of complex examples mentioned by the reviewer, we have added an example of squalane (59), which can be successfully borylated in 43% yield.*

The major concern of this research is raised from the discussion/interpretation of the unconventional selectivity and related reaction mechanism. The authors observed similar selectivity for several substrates as Aggarwal's published work (ref 64). And the authors rationalized such unconventional selectivity by steric effect only, e.g. primary over tertiary borylation. Accordingly, the authors proposed the generation of "free" chlorine radical for the following HAT. However, clearly the observed regioselectivities in the borylation indicate that "free" chlorine radicals are unlikely to be the predominant HAT species. (J. Am. Chem. Soc. 80, 4997–5001 (1958). J. Org. Chem. 63, 63, 8860–8864 (1998).)

Response: *We are very grateful for the reviewer's careful review and kind suggestion. To understand the unconventional selectivity and related reaction mechanism, several mechanistic studies have been performed, as shown below (also compiled in Fig. 5 of the revised manuscript).*

1) H/D scrambling experiment

According to existing literature reports, Cl radicals undergo HAT based on their inherent selectivity (tertiary>secondary>primary), but more substituted radicals react slower with certain electrophilic reagents, resulting in faster functionalization of primary radicals. This means that the formation of alkyl radicals is reversible (or alkyl can be "isomerized" by subsequent reversible HAT with other

molecules of the substrate) and rapid (consistent with KIE value 1). We validated this conclusion through H/D scrambling experiments. The following three deuterated compounds (**S55-d**, **S56-d**, and **S57-d**) were synthesized with moderate to high levels of deuterium incorporation at the tertiary sites, based on the method developed by Wu and colleagues (*Chem. Sci.* **2020**, *11*, 8912).

Synthesis of deuterated substrates

J. Wu, et al.
Chem. Sci. **2020**, *11*, 8912

● major deuteration sites
● minor deuteration sites

We chose **S55-d** (with 58% deuterium incorporation at the tertiary site), **S56-d** (with 93% deuterium incorporation at the tertiary site), and **S57-d** (with 85% deuterium incorporation at the tertiary site) as suitable substrates for the H/D scrambling experiment. As shown in the following scheme, substrate **S55-d** undergoes $C(sp^3)-H$ borylation reaction under the standard conditions to obtain the corresponding borylated product **55-d** in 52% yield. As expected, $C(sp^3)-H$ borylation only occurred at the terminal methyl group and significant H/D perturbations were observed, with only 23% D-incorporation at the tertiary site (35% lower than the original D-incorporation). The reaction of **S56-d** as substrate achieved similar results (36% lower than the original D-incorporation at the tertiary site), and the reaction of **S57-d** also achieved similar results (27% lower than the original D-incorporation at the tertiary site). All these results indicate that the reaction can preferentially form tertiary carbon radicals at first. Due to the slower reactivity of these more substituted radicals with certain radical acceptors, the primary carbon radicals are subsequently generated through a reversible HAT process and then react with $B_2(cat)_2$ to achieve the unconventional site selectivity.

¹H NMR spectrum (400 MHz, CDCl₃) of **S55-d**

¹H NMR spectrum (400 MHz, CDCl₃) of **55-d**

¹H NMR spectrum (400 MHz, CDCl₃) of **S56-d**

¹H NMR spectrum (400 MHz, CDCl₃) of **56-d**

¹H NMR spectrum (400 MHz, CDCl₃) of **S57-d**

¹H NMR spectrum (400 MHz, CDCl₃) of **57-d**

2) Regioselectivity studies on the reaction of *n*-pentane

i) Effect of $B_2(cat)_2$ stoichiometry

Procedure: To a 10 mL vial equipped with a stir bar was added $B_2(cat)_2$ (0.05-0.40 mmol, 0.5-2.0 equiv.), Et_4NCl (0.1 mmol, 0.5 equiv.), and HCl (concentrated, 0.8 mmol, 4.0 equiv.). $MeCN$ (6 mL) was then added followed by the addition of *n*-pentane (2.0 mmol, 10 equiv.). The reaction mixture was stirred at 680 rpm, irradiated with 390 nm LED lamps and 2.0 mA electrolysis for 12 h. The reaction temperature was maintained at approximately room temperature by cooling with a desk fan. After irradiation, a solution of pinacol (71 mg, 0.60 mmol, 3.0 equiv.) and Et_3N (0.84 mL, 6.0 mmol, 30 equiv.) in CH_2Cl_2 (1 mL) was added and stirring was continued for 1 h. Add trimethoxybenzene (internal standard) to the reaction mixture. After 1 minute of vigorous oscillation, filter 0.20 mL of the crude reaction mixture through a silica short plug and determine the yield through GC analysis.

entry	$B_2(cat)_2$ equivalents	product ratio			GC-MS yield (%)	2 % selectivity ($13\beta + 13\gamma$)/ 13α
		13α	13β	13γ		
1	0.5	73	22	5	19	0.37
2	1	54	33	13	65	0.85
3	2	37	45	19	74	1.72

ii) Effect of HCl stoichiometry

Procedure: To a 10 mL vial equipped with a stir bar was added $B_2(cat)_2$ (48mg, 0.2 mmol, 1.0 equiv.), Et_4NCl (0.1 mmol, 0.5 equiv.), and HCl (concentrated, 0.1-1.2 mmol, 0.5-6.0 equiv.). $MeCN$ (6 mL) was then added followed by the addition of *n*-pentane (2.0 mmol, 10 equiv.). The reaction mixture was stirred at 680 rpm, irradiated with 390 nm LED lamps and 2.0 mA electrolysis for 12 h. The

reaction temperature was maintained at approximately room temperature by cooling with a desk fan. After irradiation, a solution of pinacol (71 mg, 0.60 mmol, 3.0 equiv.) and Et_3N (0.84 mL, 6.0 mmol, 30 equiv.) in CH_2Cl_2 (1 mL) was added and stirring was continued for 1 h. Add trimethoxybenzene (internal standard) to the reaction mixture. After 1 minute of vigorous oscillation, filter 0.20 mL of the crude reaction mixture through a silica short plug and determine the yield through GC analysis.

entry	HCl equivalents	product ratio			GC-MS yield (%)	2 % selectivity ($13\beta + 13\gamma$)/ 13α
		13α	13β	13γ		
1	0.5	66	23	11		0.52
2	1	60	28	12	33	0.67
3	2	55	32	13	41	0.72
4	4	54	33	13	65	0.85
5	6	48	35	17	58	1.08

iii) Effect of n-pentane stoichiometry

Procedure: To a 10 mL vial equipped with a stir bar was added $\text{B}_2(\text{cat})_2$ (48mg, 0.2 mmol, 1.0 equiv.), Et_4NCl (0.1 mmol, 0.5 equiv.), and HCl (concentrated, 0.8 mmol, 4.0 equiv.). MeCN (6 mL) was then added followed by the addition of n-pentane (0.6-10.0 mmol, 3.0-50.0 equiv.). The reaction mixture was stirred at 680 rpm, irradiated with 390 nm LED lamps and 2.0 mA electrolysis for 12 h. The reaction temperature was maintained at approximately room temperature by cooling with a desk fan. After irradiation, a solution of pinacol (71 mg, 0.60 mmol, 3.0 equiv.) and Et_3N (0.84 mL, 6.0 mmol, 30 equiv.) in CH_2Cl_2 (1 mL) was added and stirring was continued for 1 h. Add trimethoxybenzene (internal standard) to the reaction mixture. After 1 minute of vigorous oscillation, filter 0.20 mL of the crude reaction mixture through a silica short plug and determine the yield through GC analysis.

entry	pentane equivalents	product ratio			GC-MS yield (%) ^a	2 % selectivity (26β + 26γ)/26α
		26α	26β	26γ		
1	3	61	29	10	21	0.64
2	5	56	32	12	47	0.78
3	10	54	33	13	63	0.85
4	20	46	39	15	65	1.17
5	50	44	40	16	59	1.27

3) Regioselectivity studies on the reaction of 2,3-dimethylbutane (DMB)

Since the chlorine radical-mediated HAT step has proved to be not turnover-limiting through our KIE experiment, we hypothesized that the afforded regioselectivity is highly dependent on the choice of radical acceptors. To testify our hypothesis, we chose 2,3-dimethylbutane (DMB), a branched isomer of hexane that was recently utilized to probe the HAT species by Zuo (*J. Am. Chem. Soc.* **2020**, *142*, 6216–6226; *J. Am. Chem. Soc.* **2023**, *145*, 359–376) and Walsh/Schelter (*Science* **2021**, *372*, 847–852), as a standard hydrocarbon substrate for the evaluation of site selectivity. The two methine positions of DMB contain weaker 3° C–H bonds in comparison to the four distal methyl groups. Several radical acceptors have been tested under our photoelectrochemical conditions as well as other chlorine radical-mediated photochemical systems (ref. a-c). All these results indicate that the chlorine radical-mediated HAT process preferentially occur at the weaker 3° C–H bonds according to its intrinsic selectivity. However, the site selectivity is also largely affected by the use of radical acceptors.

Procedure: To a 10 mL vial equipped with a stir bar was add free radical acceptor (0.20 mmol, 1.0 equivalent), Et_4NCl (0.1 mmol, 0.5 equivalent), and HCl (concentrated, 0.8 mmol, 4.0 equivalent) to a 10 mL vial containing a stirring rod. Then add MeCN (6 mL), and then add 2,3-dimethylbutane (2.0 mmol, 10 equivalents). The reaction mixture was stirred at 680 rpm, illuminated with a 390 nm LED lamp, and electrolyzed at 2.0 mA for 12 hours. Maintain the reaction temperature at approximately room temperature by cooling with a desktop fan. After irradiation, the reaction mixture was vacuum concentrated and purified through rapid column chromatography to obtain the corresponding transparent oil like product. The intrinsic selectivity of 1°/3° $\text{C}(\text{sp}^3)\text{-H}$ bonds was determined by ^1H NMR analysis.

HAT condition	radical acceptor	intrinsic selectivity (1°:3°)	yield (%)
	$\text{B}_2(\text{cat})_2$	> 20:1	72
	<chem>C=CC(=O)OPh</chem>	1:6	74
	<chem>C=C(C#N)C#N</chem>	1:3	70
$\text{FeCl}_3 \cdot 6\text{H}_2\text{O}^{\text{a}}$	<chem>BocN=N(Boc)C</chem>	1:5	84
$\text{Mes-Acr}^+/\text{nBu}_4\text{NCl}^{\text{b}}$	<chem>BocN=N(Boc)C</chem>	1:6	26
$\text{FeCl}_3 \cdot 6\text{H}_2\text{O}^{\text{c}}$	<chem>p-TolC#CC(=O)OPh</chem>	> 99:1	62

a) Duan, C. et al. *Green Chem.* **2021**, 23, 6984-6989.

b) Zuo, Z. et al. *J. Am. Chem. Soc.* **2023**, 145, 359-376.

c) Duan, C. et al. *Green Chem.* **2021**, 23, 9406-9411.

Under the standard photoelectrochemical conditions of our $\text{C}(\text{sp}^3)\text{-H}$ borylation reaction, control reaction revealed that the addition of benzylidenemalononitrile (**64**) as a typical radical acceptor greatly inhibited the desired C-H borylation reaction, while generating C-H alkylated product **65** preferentially at the weaker 3° C-H bond (intrinsic selectivity of 1°/3° 1:3). Consistent with this finding, the addition of phenyl acrylate (**66**) into the photoelectrochemical system with DMB (**S9**) and $\text{B}_2(\text{cat})_2$ (**2**) produced C-H alkylated product **67** in 65% yield, and again, the alkylation at the 3° C-H bond is preferred (intrinsic selectivity of 1°/3° 1:6). These results supported the existence of tertiary carbon radicals as well as the reversible HAT process, which is consistent with the H/D scrambling results.

Procedure: To a 10 mL vial equipped with a stir bar was add free radical acceptor (0.20 mmol, 1.0 equivalent), $\text{B}_2(\text{cat})_2$ (48 mg, 0.20 mmol, 1.0 equiv.), Et_4NCl (0.1 mmol, 0.5 equivalent), and HCl (concentrated, 0.8 mmol, 4.0 equivalent) to a 10 mL vial containing a stirring rod. Then add MeCN (6 mL), and then add 2,3-dimethylbutane (2.0 mmol, 10 equivalents). The reaction mixture was

stirred at 680 rpm, illuminated with a 390 nm LED lamp, and electrolyzed at 2.0 mA for 12 hours. Maintain the reaction temperature at approximately room temperature by cooling with a desktop fan. After irradiation, a solution of pinacol (71 mg, 0.60 mmol, 3.0 equiv.) and Et_3N (0.84 mL, 6.0 mmol, 20 equiv.) in CH_2Cl_2 (1 mL) was added and stirring was continued for 1 h. The intrinsic selectivity of $1^\circ/3^\circ\text{C}(\text{sp}^3)\text{-H}$ bonds was determined by ^1H NMR analysis.

i) Selective study on the reaction of 2,3-dimethylbutane with 2-benzylidenemalononitrile (**64**): Alkyl boronate ester **9** was formed in 24% yield with complete terminal 1°C-H selectivity. The alkylated product **65** was formed in 43% yield and 2:1 ratio of 1° and 3° alkylated products (intrinsic selectivity of $1^\circ/3^\circ = 1:3$).

ii) Selective study on the reaction of 2,3-dimethylbutane with phenyl acrylate (**66**): Alkyl boronate ester **9** was formed in 7% yield with complete terminal 1°C-H selectivity. The alkylated product **67** was formed in 65% yield and 1:1 ratio of 1° and 3° alkylated products (intrinsic selectivity of $1^\circ/3^\circ = 1:6$).

4) Electron paramagnetic resonance (EPR) studies

An electron paramagnetic resonance (EPR) experiment has been performed to detect key radical intermediates by adding a free radical spin trap 5,5-dimethyl-1-pyrroline *N*-oxide (DMPO). From the afforded EPR spectroscopy (as shown below) it is suggested that carbon radicals are generated and rapidly captured by DMPO to form relatively stable free radicals ($g = 2.007$, $A_N = 14.81$ G, and $A_{\text{H}\beta} = 21.38$ G). All the above mechanistic experiments supported the engagement of chlorine radical species and formation of carbon-centered radicals via intermolecular HAT process in the photoelectrochemical $\text{C}(\text{sp}^3)\text{-H}$ borylation reaction.

Procedure: To a 10 mL vial equipped with a stir bar was added $\text{B}_2(\text{cat})_2$ (48 mg, 0.2 mmol, 1.0 equiv.), Et_4NCl (0.1 mmol, 0.5 equiv.), and HCl (concentrated, 0.8 mmol, 4.0 equiv.). MeCN (3 mL) was then added followed by the pentane (2.0 mmol, 10 equiv.). The reaction mixture was stirred at 680 rpm, irradiated with 390 nm LED lamps and 2.0 mA electrolysis. The reaction temperature was maintained at approximately room temperature by cooling with a desk fan, using a constant current for 0.5 h. Add DMPO (15 mmol) to the reaction bottle and continue stirring for 10min, and

afterwards the solution sample was taken out into a small tube for EPR test. EPR spectra was recorded at room temperature on EPR Elexsys E500 spectrometer operated at 9.855 GHz. Typical spectrometer parameters were shown as follows, sweep width: 200.00 G; center field set: 3500.00 G; conversion time: 58.59 ms; sweep time: 59.99 s; modulation amplitude: 1.0 G; modulation frequency: 100 kHz; PowerAtten10.0 dB; microwave power: 20.00 mW. We performed the EPR experiment to detect radical intermediates by adding a free radical spin trap DMPO. It is suggested that carbon radicals are formed and rapidly captured by DMPO to form relatively stable free radicals ($g = 2.007$, $A_N = 14.81$, and $A_{H\beta} = 21.38$).

Types of free radicals	gFactor	Hyperfine coupling constant			Content (%)
		A_N	$A_{H\beta}$	$A_{H\gamma}$	
DMPO-R	2.007	14.8197	21.3883	-	95.71
DMPO-X		13.7637	-	-	4.29

5) Explanation of HAT species

To gain more insight into the origination of HAT species, styrene (**60**) was added into the photoelectrochemical system and a chlorine-containing ketone product **61** that captured chlorine radical species at the terminal alkene was successfully isolated in 69% yield, indicating the generation and involvement of chlorine radicals during our reaction process. Consistent with this finding, the electrolysis of a solution of *N,N*-diallyltosylamine (**62**), HCl, and Et_4NCl in CH_3CN produced the chlorinated ring-closing product **63** derived from a chlorine radical addition–5-exo-trig cyclization–hydrogenation sequence (for the last hydrogenation step in electrochemical system, please see a related reference: *Angew. Chem. Int. Ed.* **2016**, *55*, 2226–2229). However, in the

presence of $B_2(\text{cat})_2$, the formation of borylated compound **63'** via chlorine radical addition–5-exo-trig cyclization–borylation sequence was not detected. All these results suggest the involvement of chlorine radical species in the reaction process.

Procedure: To a 10 mL vial equipped with a C anode ($\varphi = 3$ mm), a platinum plate cathode (10 x 10 x 0.1 mm), and a magnetic stir bar was added with styrene (**60**, 0.3 mmol, 1 equiv). After three cycles of evacuation and backfilling of the reaction flask with argon, a solution of Et_4NCl (0.15 mmol, 0.5 equiv) in CH_3CN (6 mL) and HCl (concentrated, 1.2 mmol, 4 equiv) were added. The electrolysis was carried out in dark using a constant current of 2 mA at room temperature until complete consumption of styrene. The reaction mixture was concentrated in vacuo and purified by flash column chromatography. The target compound was obtained with a yield of 69%. All recorded spectroscopic data matched those previously reported in the literatures.

2-Chloro-1-phenylethan-1-one (**61**): $^1\text{H NMR}$ (400 MHz, chloroform-*d*) δ 7.95 (d, $J = 7.9$ Hz, 2H), 7.61 (t, $J = 7.4$ Hz, 1H), 7.49 (t, $J = 6.9$ Hz, 2H), 4.72 (s, 2H). $^{13}\text{C NMR}$ (101 MHz, chloroform-*d*) δ 191.2, 134.3, 134.1, 129.0, 128.6, 46.2.

Procedure: To a 10 mL vial equipped with a C anode ($\varphi = 3$ mm), a platinum plate cathode (10 x 10 x 0.1 mm), and a magnetic stir bar was added with *N,N*-diallyl-*N*-tosylamine (**62**, 0.3 mmol, 1 equiv). After three cycles of evacuation and backfilling of the reaction flask with argon, a solution of Et_4NCl (0.15 mmol, 0.5 equiv) in CH_3CN (6 mL) and HCl (concentrated, 1.2 mmol, 4 equiv) were added. The electrolysis was carried out in dark using a constant current of 2 mA at room temperature until complete consumption of compound **62**. The reaction mixture was concentrated in vacuo and purified by flash column chromatography. The target compound was obtained with a yield of 57%.

3-(chloromethyl)-4-methyl-1-tosylpyrrolidine (**63**): ^1H NMR (400 MHz, chloroform-*d*) δ 7.69 (dt, $J = 8.1, 2.2$ Hz, 2H), 7.32 (d, $J = 7.9$ Hz, 2H), 3.55 – 3.34 (m, 3H), 3.35 – 3.25 (m, 0.42 H), 3.23 – 3.09 (m, 1.58H), 2.99 (ddd, $J = 9.8, 4.9, 1.6$ Hz, 0.66 H), 2.83 – 2.75 (m, 0.39 H), 2.47 – 2.25 (m, 4.52 H), 1.97 (dp, $J = 20.2, 7.4, 6.2$ Hz, 0.76 H), 0.94 (dd, $J = 6.3, 1.7$ Hz, 1H), 0.81 (dd, $J = 6.9, 1.6$ Hz, 2H). ^{13}C NMR (101 MHz, chloroform-*d*) δ 143.7, 143.6, 133.6, 133.3, 129.8, 127.6, 127.5, 54.8, 54.4, 51.4, 50.1, 47.7, 44.9, 44.4, 42.8, 36.4, 34.9, 21.6, 16.6, 12.8.

According to the reviewer's comments, we agree with the reviewer's opinion for the HAT species. The chlorine radical trapping experiment cannot prove that the "free" Cl radical is the HAT species, as the Cl radical may react with $\text{B}_2(\text{cat})_2$ to form a reactive chlorine-containing complex (just like the radical 'ate' complex proposed in Aggarwal's work: *Nature* **2020**, 586, 714–719), which further undergoes the HAT process with hydrocarbons. We also tried very hard to isolate possible reaction intermediates to probe the active HAT species, but unfortunately all our trials failed in this regard, probably due to the fast and reversible HAT process in our photoelectrochemical system. Considering the above-mentioned mechanistic experiments including KIE experiment, H/D scrambling experiment and radical trapping with alkenes, we believe that clear evidence has been provided to explain the afforded unconventional regioselectivity.

Furthermore, the references mentioned by the reviewer have been cited: *J. Am. Chem. Soc.* **1958**, 80, 4997–5001 as ref. 91; *J. Org. Chem.* **1998**, 63, 8860–8864 as ref. 92.

Is it possible the radical chain mechanism is propagated by the reaction of chloride-stabilised boryl radical as the same or similar intermediate proposed by Aggarwal (ref 64). Obviously, more mechanistic studies should be conducted. For example, the authors could refer to Aggarwal's published work (ref 64).

Response: We are very grateful for the reviewer's careful review and good suggestion. It is possible to utilize the commercially available $\text{ClB}(\text{cat})$ as the chlorine source and initiate a radical chain process. Although the reaction with $\text{ClB}(\text{cat})$ instead of HCl gave no desired borylated product in an undivided-cell reaction, it works in a divide cell to deliver product **3** in 8% yield with the use of 50 mol% $\text{ClB}(\text{cat})$ instead of HCl, and Et_4NBF_4 instead of Et_4NCl .

Procedure: To a 10 mL divide cell equipped with a stir bar, in anodic chamber was added $\text{B}_2(\text{cat})_2$ (48 mg, 0.20 mmol, 1.0 equiv.), Et_4NBF_4 (0.10 mmol, 0.5 equiv.), and ClBcat (0.10 mmol, 0.5 equiv.)

in MeCN (6 mL) followed by cyclohexane (2.0 mmol, 10 equiv.). In cathodic chamber was added Et_4NBF_4 (0.10 mmol, 0.5 equiv.) and MeCN (6 mL). The reaction mixture was stirred at 680 rpm, irradiated with 390 nm LED lamps and 2.0 mA for only 2 h, and a reaction sample was picked up for the post-processing of GC analysis. The yield of the target product **3** was found to be 8%. After 8 h irradiation and electrolysis, the yield did not increase.

As for the SI, more GC spectra should be provided to clarify the unconventional selectivity. More than 20 substrates showed the unconventional selectivity, however, only 8 examples are provided with GC spectra.

Response: *We are very grateful for the reviewer's careful review and kind suggestion. We have added the corresponding GC spectra into the revised SI.*

Reviewer #2

The authors present an interesting approach to the oxidative C(sp³)-H borylation of hydrocarbons using photoelectrochemistry, eliminating the need for metals and oxidants. Organoboron compounds hold immense potential in organic synthesis, and the development of an efficient and versatile method for their synthesis is of high importance. One of the primary strengths of this study is the broad substrate scope and interesting regioselectivity of C(sp³)-H borylation, with a preference for the distal methyl position. The scalability of the procedure makes it even more appealing, as it facilitates access to high-value organoboron building blocks from simple hydrocarbons in a single step. I recommend accepting this article for publication after minor revisions. The authors should address the following points:

1. Provide more details on the mechanism of radical borylation and how it leads to the observed regioselectivity. The HAT process with chlorine atom is expected to be much less selective. Likely, only selected carbon radicals made their way to the organoboron products.

Response: *We are very grateful for the reviewer's careful review and kind suggestion. We have added very detailed mechanistic studies and in-depth understanding of the afforded site selectivity into the revised manuscript (please see Fig. 5 in the revised manuscript as well as the answer to Reviewer #1).*

2. Add results with less hydrocarbons to Table 1. The use of large excess of hydrocarbons is a major limitation although simple alkanes can be recovered.

Response: *We are very grateful for the reviewer's careful review and good suggestion. We have added the corresponding results with less hydrocarbons in the revised Table 1 (entry 14). The yield*

of alkyl boronate **3** slightly decreased (58%) when the reaction was performed using 5.0 equiv. of cyclohexane **1**.

3. Explain the drastically different results with $n\text{Bu}_4\text{NCl}$ and Et_4NCl .

Response: We are very grateful for the reviewer's careful review and kind suggestion. The use of $n\text{Bu}_4\text{NCl}$ instead of Et_4NCl would promote the hydrolysis of $\text{B}_2(\text{cat})_2$ as well as other side reaction process. Following the advice, we have conducted some research to investigate the origin of the drastic difference on yield with the use of $n\text{Bu}_4\text{NCl}$ and Et_4NCl . When using $n\text{Bu}_4\text{NCl}$ to replace Et_4NCl for reaction, no target borylated product was detected. We have found significant differences in the ^1H NMR spectra obtained by studying two different electrolytes. When using Et_4NCl for the photoelectrochemical reaction, the ^1H NMR spectrum was very clean and no obvious by-products were detected. However, when $n\text{Bu}_4\text{NCl}$ was used for the reaction, the ^1H NMR spectrum showed the formation of hydrolysis product (catechol) of B_2cat_2 together with some other unknown by-products. We have added the relevant experimental results to the revised SI.

i) Fig. 1 shows the ^1H NMR spectra of $\text{B}_2(\text{cat})_2$ and catechol in CDCl_3

Fig. 1 ^1H NMR spectra of $\text{B}_2(\text{cat})_2$ and catechol in CDCl_3

ii) NMR spectra of reaction mixtures with Et_4NCl or $n\text{Bu}_4\text{NCl}$:

To two 10 mL vials equipped with stir bars was added $\text{B}_2(\text{cat})_2$ (48mg, 0.2 mmol, 1.0 equiv.), Et_4NCl or $n\text{Bu}_4\text{NCl}$ (0.1 mmol, 0.5 equiv.), and HCl (concentrated, 0.8 mmol, 4.0 equiv.). MeCN (6 mL) was then added followed by the pentane (2.0 mmol, 10 equiv.). The reaction mixture was stirred at 680 rpm, irradiated with 390 nm LED lamps and 2.0 mA electrolysis. The reaction temperature was maintained at approximately room temperature by cooling with a desk fan. Before the start of the reaction, samples were picked up from these two vials for NMR analysis (time = 0 h, see Fig. 2).

After the reaction mixture was stirred at 680 rpm and irradiated with 390 nm LED lamps for 8 h, irradiation was stopped and samples were picked up from these two vials for NMR analysis (time = 8 h, see Fig. 3).

Fig. 2. ¹H NMR spectra of reaction mixtures (time = 0 h): a) Et₄NCl; b) nBu₄NCl.

Fig. 3. ¹H NMR spectra of reaction mixtures (time = 8 h): a) Et₄NCl; b) nBu₄NCl.

4. CF₃OH or CF₃CH₂OH?

Response: We are very grateful for the reviewer's careful review and kind suggestion. When CF₃OH is used as solvent, the yield of alkylboron **3** is 39%. The use of CF₃CH₂OH as solvent delivered alkylboron **3** in 27% yield. We have added these results in the revised Table 1 (entries 10-11).

5. ClB(cat) was important for Aggarwal's work. It is important here?

Response: We are very grateful for the reviewer's careful review and good suggestion. It is possible to utilize the commercially available ClB(cat) as the chlorine source and initiate a radical chain process. Although the reaction with ClB(cat) instead of HCl gave no desired borylated product in an undivided-cell reaction, it works in a divide cell to deliver product **3** in 8% yield with the use of 50 mol% ClB(cat) instead of HCl, and Et₄NBF₄ instead of Et₄NCl. However, considering the high price of ClB(cat), our photoelectrochemical method shows greater advantages.

Procedure: To a 10 mL divide cell equipped with a stir bar, in anodic chamber was added B₂(cat)₂ (48 mg, 0.20 mmol, 1.0 equiv.), Et₄NBF₄ (0.10 mmol, 0.5 equiv.), and ClBcat (0.10 mmol, 0.5 equiv.) in MeCN (6 mL) followed by cyclohexane (2.0 mmol, 10 equiv.). In cathodic chamber was added Et₄NBF₄ (0.10 mmol, 0.5 equiv.) and MeCN (6 mL). The reaction mixture was stirred at 680 rpm, irradiated with 390 nm LED lamps and 2.0 mA for only 2 h, and a reaction sample was picked up for the post-processing of GC analysis. The yield of the target product **3** was found to be 8%. After 8 h irradiation and electrolysis, the yield did not increase.

6. There is no evidence that **57** is formed through addition of a chlorine atom to styrene.

Response: We are very grateful for the reviewer's careful review and kind suggestion. We have added another chlorine radical-trapping experiment in order to illustrate the involvement of Cl radical species in the reaction process. The electrolysis of a solution of *N,N*-diallyltosylamine (**62**), HCl, and Et₄NCl in CH₃CN produced the chlorinated ring-closing product **63** derived from a chlorine radical addition–5-exo-trig cyclization–hydrogenation sequence (for the last hydrogenation step in electrochemical system, please see a related reference: *Angew. Chem. Int. Ed.* **2016**, *55*, 2226–2229). However, in the presence of B₂(cat)₂, the formation of borylated compound **63'** via chlorine radical addition–5-exo-trig cyclization–borylation sequence was not detected.

Procedure: To a 10 mL vial equipped with a C anode ($\varphi = 3$ mm), a platinum plate cathode (10 x 10 x 0.1 mm), and a magnetic stir bar was added with *N,N*-diallyltosylamine (**62**, 0.3 mmol, 1 equiv). After three cycles of evacuation and backfilling of the reaction flask with argon, a solution of Et_4NCl (0.15 mmol, 0.5 equiv) in CH_3CN (6 mL) and HCl (concentrated, 1.2 mmol, 4 equiv) were added. The electrolysis was carried out in dark using a constant current of 2 mA at room temperature until complete consumption of compound **62**. The reaction mixture was concentrated in vacuo and purified by flash column chromatography. The target compound was obtained with a yield of 57%.

3-(chloromethyl)-4-methyl-1-tosylpyrrolidine (**63**): $^1\text{H NMR}$ (400 MHz, chloroform-*d*) δ 7.69 (dt, $J = 8.1, 2.2$ Hz, 2H), 7.32 (d, $J = 7.9$ Hz, 2H), 3.55 – 3.34 (m, 3H), 3.35 – 3.25 (m, 0.42 H), 3.23 – 3.09 (m, 1.58H), 2.99 (ddd, $J = 9.8, 4.9, 1.6$ Hz, 0.66 H), 2.83 – 2.75 (m, 0.39 H), 2.47 – 2.25 (m, 4.52 H), 1.97 (dp, $J = 20.2, 7.4, 6.2$ Hz, 0.76 H), 0.94 (dd, $J = 6.3, 1.7$ Hz, 1H), 0.81 (dd, $J = 6.9, 1.6$ Hz, 2H). $^{13}\text{C NMR}$ (101 MHz, chloroform-*d*) δ 143.7, 143.6, 133.6, 133.3, 129.8, 127.6, 127.5, 54.8, 54.4, 51.4, 50.1, 47.7, 44.9, 44.4, 42.8, 36.4, 34.9, 21.6, 16.6, 12.8.

7. What are the limitations?

Response: The limitation of this photoelectrochemical protocol is that the reaction cannot be well applied to nitrogen-containing hydrocarbon substrates. For example, we have tested amides, amines, and sulfonamides under the standard conditions, but the reaction of using these compounds as substrates cannot yield the desired C–H borylated products. We have added one sentence to describe this limitation in the revised manuscript.

8. 390 nm LED is purple LED not blue LED.

Response: We are very grateful for the reviewer's careful review and kind suggestion. We have corrected this error.

9. It is inaccurate to describe tetraethylammonium chloride as electrolyte and HCl as the chlorine source. They both provide chloride and increase conductivity. HCl is needed likely to increase proton reduction to avoid reduction of the boron reagent. What if you use other acids, such as TFA, TfOH.

Response: We gratefully acknowledge the reviewer for this very constructive suggestion, which undoubtedly facilitates the explanation of the reaction mechanism. In order to further understand the role of HCl in this photoelectrochemical system, several control experiments were carried out, as shown in Fig. 5g of the revised manuscript. Replacing HCl with other inorganic chloride salts, such as NaCl, CsCl, and LiCl, was found to be ineffective. Although the use of other acids instead of HCl, such as TFA and TfOH, led to complete loss of reaction efficiency (Table 1, entry 15), substituting HCl by H₃PO₄ successfully gave the desired C–H borylated product **3** in 67% yield, which indicates that HCl mainly plays a crucial role as proton source rather than chlorine source. At the cathode surface, protons undergo cathodic reduction to generate H₂, and protect the substrate B₂(cat)₂ from single-electron reduction in certain conditions (*J. Am. Chem. Soc.* **2021**, *143*, 12985–12991).

Entry	Deviation from standard conditions	Yield (%)
1	None	72
2	NaCl / CsCl / LiCl instead of HCl	< 5
3	H ₃ PO ₄ instead of HCl	67

Moreover, a divided-cell experiment was carried out. Substrates **1** and **2**, along with Et₄NCl, HCl, and CH₃CN, was placed into the anode chamber, while Et₄NCl and CH₃CN were added into the cathode chamber. As expected, upon light irradiation and electrolysis for 12 h the desired product **3** was detected in the anode chamber with 46% yield, suggesting that the photoelectrochemical C(sp³)-H borylation occurred surrounding the graphite rod anode. Interestingly, a moderate yield (41%) of alkyl boronate **3** was still afforded in the absence of HCl at the anode chamber, probably because the B₂(cat)₂ reagent in the anode chamber of divided-cell would not be electrochemically reduced, and therefore it does not need the protection of HCl any more. The protons that come from trace amount of H₂O in the cathode cell are combined with electrons at the cathode to generate H₂.

Entry	Deviation from standard conditions	Yield (%)
1	None	46
2	Without HCl at anode	41

10. Please give more details on the components of the photoelectrochemical cell to allow others to reproduce the results.

Response: We are very grateful for the reviewer's careful review and kind suggestion. More details about the photochemical equipment and electrochemical cell have been added to the revised SI.

Reviewer #3

The Manuscript by Xia, Guo and co-authors describes a terminal oxidant and transition metal-free, photoelectrochemical C(sp³)-H borylation. This method involves intermolecular hydrogen atom transfer by the chlorine radical, generated via anodic oxidation of chloride, and followed by photoinduced homolysis of the produced chlorine gas. The resulting alkyl radical was trapped with diboron reagent to afford alkyl boronates. Expectedly, this method exhibits high selectivity towards terminal C-H sites. This operationally simple and scalable protocol can be employed for broad range of substrates including halides, silanes, ketones and esters.

I think this work does not represent significant advances to meet the high standard of Nature Communications for the following reasons. This method involves known photoelectrochemical strategy (Ref: 14) for radical generation. It exhibits alike site-selectivity, as in previous radical C-H borylation reports (Ref: 64), and in the authors' recent work, which is not cited in this manuscript (J. Am. Chem. Soc. 2023, 145, 7600–7611).

Response: We are very grateful for the reviewer's careful review and kind suggestion. Our recent work about C-H borylation reaction (J. Am. Chem. Soc. 2023, 145, 7600–7611) has been cited in the revised manuscript as ref. 65. In comparison to our previous work using iron catalyst and NFSI as oxidant, the newly reported photoelectrochemical protocol is metal free (the Pt cathode does not participate in the reaction, and can be replaced by a graphite rod electrode with slight decrease of

reaction efficiency) and sustainable, which is more welcomed by the pharmaceutical industries as it is mandatory to remove toxic trace metal contaminants from products. Gram-scale synthesis of alkylboron product has been performed on a 20 mmol scale in continuous-flow, which greatly highlights the synthetic potential of our method. Furthermore, very detailed mechanistic studies and in-depth understanding of the afforded site selectivity have been added into the revised manuscript, including the EPR measurement, H/D scrambling experiment, radical trapping reactions, as well as the investigations of the HAT species. Based on this newly added mechanistic results, we believe that this protocol has exhibited some novel aspect compared to Aggarwal's and our previous C–H borylation reaction (ref. 64 & 65) and Xu's photoelectrochemical C–H arylation protocol (ref. 14). Therefore we really appreciate it if you could reconsider our new submission.

Reviewers' comments:

Reviewer #1 (Remarks to the Author):

This is a second review of the submitted manuscript with revisions. After carefully reviewing the authors' response and additional experiments, the reviewer does not think this manuscript represents significant advances in methodology/synthesis to meet the high standard of Nature Communications for the following reasons.

1. To address the reviewer's comments, the authors provided more detailed mechanistic studies. However, these studies shown in Figure 5 only provide evidence that the HAT process at 1/2/3 sites is reversible and the selectivity is radical acceptor depended, which is well known. These studies did not answer the reviewer's question: what is the HAT species? Is it free Cl radical or Cl-boron complexes? Is the unconventional selectivity only determined by the reversible HAT process/ radical acceptor, the HAT species, or both?
2. From a methodology aspect, this research involves a known photoelectrochemical strategy (Ref: 14) for radicals generation. Moreover, the unconventional selectivity for radical borylation has been reported in several reports (e.g. ref 64,65). It is not significant to the field and related fields, unless the above-mentioned questions in 1 could be answered. Otherwise, this investigation is just an synthetic extension of ref 14.

Reviewer #2 (Remarks to the Author):

The authors have addressed previous comments. To ensure replicability of the outcomes, it is crucial to furnish comprehensive details about the flow reactor, including its design, operating conditions, and specific parameters. This will facilitate an accurate recreation of the experiment in different settings.

Point-by-point response

Reviewer #1 (Remarks to the Author):

This is a second review of the submitted manuscript with revisions. After carefully reviewing the authors' response and additional experiments, the reviewer does not think this manuscript represents significant advances in methodology/synthesis to meet the high standard of Nature Communications for the following reasons.

1. To address the reviewer's comments, the authors provided more detailed mechanistic studies. However, these studies shown in Figure 5 only provide evidence that the HAT process at 1/2/3 sites is reversible and the selectivity is radical acceptor depended, which is well known. These studies did not answer the reviewer's question: what is the HAT species? Is it free Cl radical or Cl-boron complexes? Is the unconventional selectivity only determined by the reversible HAT process/ radical acceptor, the HAT species, or both?

Response: We are very grateful for the reviewer's comments. This viewpoint of depended radical acceptor is actually not well known, which is only mentioned in our previous research paper (*J. Am. Chem. Soc.* **2023**, *145*, 7600–7611) without giving very solid evidence. In this photoelectrochemical work, similar site selectivity of C(sp³)-H borylation is observed with reliable evidence provided by Cl radical trapping reactions, EPR spectroscopy, H/D scrambling, trapping experiments with other radical acceptors as well as divided-cell experiment (see Fig. 5), which is a novel feature and important complement to the studies in this research area.

We really appreciate that the reviewer has raised the issue of HAT species. More mechanistic experiments have been carried out to investigate the HAT species in combination with Aggarwal's recent paper (*J. Am. Chem. Soc.* **2023**, *145*, 15207–15217). Our experimental results indicate that free Cl radical and Cl-boron complexes are both likely to be the HAT species during the reaction process.

1) Evidence for free Cl radical as HAT species

To probe whether there is a direct HAT process of free chlorine radical in the photoelectrochemical reaction, we used other type of radical receptors instead of B₂cat₂ under the standard conditions. The use of benzylidenemalononitrile (**64**) and phenyl acrylate (**66**) as typical radical acceptors successfully gave the C-H alkylated products **65** and **67** in 70% and 74% yields, respectively. Consistent with other reactions via free Cl radical HAT process, the regioselectivities of these two reactions follow the trend of 3° > 2° > 1°, giving the desired products with 1:3 and 1:6 ratio of intrinsic selectivity (1/3), respectively. If B₂cat₂ is also added into the photoelectrochemical system, a Cl-boron complex may be rapidly in situ generated and affects the site selectivity of C-H alkylation due to the sterically hindered feature of the Cl-boron complex in HAT process (see Figure 7c & 7d in Aggarwal's paper: *J. Am. Chem. Soc.* **2023**, *145*, 15207–15217). In the presence of B₂cat₂, control reaction revealed that the addition of benzylidenemalononitrile (**64**) as a typical radical acceptor greatly inhibited the desired C-H borylation reaction, while generating C-H alkylated product **65** preferentially at the weaker 3° C-H bond (the intrinsic selectivity of 1/3 is still 1:3). Consistent with this finding, the addition of phenyl acrylate (**66**) into the photoelectrochemical

system with DMB (**S9**) and B_2cat_2 (**2**) produced C–H alkylated product **67** in 65% yield with the same intrinsic selectivity of 1°:3° (1:6) as the reaction in the absence of B_2cat_2 . These results suggested that free Cl radical HAT process could be much faster than the in situ generation of a Cl-boron complex under our photoelectrochemical conditions. Therefore, unlike Aggarwal’s copper-mediated $C(sp^3)$ –H borylation protocol in which the presence of copper species greatly facilitates the generation of a Cl-boron complex, under our photoelectrochemical conditions the free Cl radical HAT may dominate the HAT-mediated functionalization reactions.

2) Evidence for Cl-boron complexes as HAT species

As mentioned in Aggarwal’s recent paper (*J. Am. Chem. Soc.* **2023**, *145*, 15207–15217), a key reaction intermediate, $O(Bcat)_2$, as a novel electrophilic borylating agent is proposed to react with free Cl radical and generate Cl-radical-boron “ate” complex, which is supposed to be the HAT species in that research work. The $O(Bcat)_2$ species is derived from the copper-catalyzed reaction of B_2cat_2 with H_2O (see the figure below).

Aggarwal's Cl-boron complex-mediated HAT process

Under our photoelectrochemical conditions, we also tried hard to capture these key intermediates, $\text{O}(\text{Bcat})_2$ and HO-Bcat , in order to get evidence of Cl-boron complex as HAT species. In the presence of HCl (4 equiv.) and Et_4NCl (0.5 equiv.), weak signals of HO-Bcat in ^1H NMR spectra are likely to be observed (see Fig. S9). In comparison to Aggarwal's results where clear NMR signals of $\text{O}(\text{Bcat})_2$ and HO-Bcat are successfully captured, we only observed weak signals of HO-Bcat probably due to the fact that the presence of metal species plays a key role in the cleavage of boron-boron bond and promotes the formation of $\text{O}(\text{Bcat})_2$ species (also see: *J. Chem. Soc., Dalton Trans.* **1998**, 301–309).

Procedure: To three 10 mL vials equipped with stir bars was added $\text{B}_2(\text{cat})_2$ (48mg, 0.2 mmol, 1.0 equiv.), Et_4NCl (0.1 mmol, 0.5 equiv.), and HCl (concentrated, 0.8 mmol, 4.0 equiv.) in the glovebox. MeCN (6 mL) was then added to the reaction system followed by the addition of cyclohexane (2.0 mmol, 10 equiv.). The reaction mixture was stirred at 680 rpm for one hour and then directly take samples for NMR analysis. Figure S9 shows that a small part of $\text{B}_2(\text{cat})_2$ was hydrolyzed into HO-Bcat by ^1H NMR detection.

Fig. S9 ^1H NMR spectra of reaction mixtures (400 MHz, Chloroform-*d*)

To verify whether a Cl-boron complex can be generated in the photoelectrochemical system, a divided-cell experiment was carried out, as shown in the figure below. Substrates (**1**) and B₂cat₂ (**2**), along with Et₄NBF₄, ClBcat, and CH₃CN, was placed into the anode chamber, while Et₄NBF₄ and CH₃CN were added into the cathode chamber. As expected, upon light irradiation and electrolysis for 12 h the desired alkyl boronate **3** was detected in the anode chamber with 8% yield, suggesting that the photoelectrochemical C(sp³)–H borylation occurred surrounding the graphite rod anode. Interestingly, an increased yield (24%) of alkyl boronate **3** was afforded in the presence of H₂O (5 μL) at the anode chamber, probably because the presence of H₂O promotes the generation of HO-Bcat species from Cl-Bcat, and HO-Bcat is the key precursor for the production of Cl-boron complex (Path A; generation of Cl-boron complex from HO-Bcat and Cl radical; Path B; generation of Cl-boron complex from O(Bcat)₂ and Cl radical).

Entry	Deviation from standard conditions	Yield (%)
1	None	8
2	Adding H ₂ O (5 μL)	24

Another evidence for the Cl-boron complex as HAT species is afforded by using squalane (**S59**) as substrate. The reaction of squalane (**S59**) with B_2cat_2 (**2**) under the standard photoelectrochemical conditions successfully delivered the desired $C(sp^3)$ -H borylation product **59** in 43% yield with exclusive site selectivity. In the meantime we were also able to isolate a terminal alkene **59'** as a major side product, which is likely to be generated from 1° radical species followed by anodic oxidation and E1 elimination (see the figure below), while the generation of internal alkene **59''** was not observed. Considering the fact that Cl radical preferentially undergoes HAT at the 3° $C(sp^3)$ -H bonds and a Cl-boron complex preferentially undergoes HAT at the sterically unhindered 1° $C(sp^3)$ -H bonds, the formation of this terminal alkene **59'** proves the existence of Cl-boron complex as HAT species in our photoelectrochemical system.

2,6,10,15,19,23-hexamethyltetracos-1-ene (**59'**)

1H NMR (400 MHz, Chloroform-*d*) δ 6.82-6.70 (m, 2H), 1.58 (s, 3H), 1.52 (td, $J = 6.7, 2.4$ Hz, 2H), 1.33 – 1.08 (m, 35H), 0.86-0.82 (m, 18H).

^{13}C NMR (101 MHz, Chloroform-*d*) δ 121.2, 108.6, 39.5, 37.5, 37.4, 32.9, 28.1, 27.6, 26.0, 24.9, 24.6, 22.9, 22.8, 19.9.

¹H NMR spectrum (400 MHz, CDCl₃)

¹³C NMR spectrum (101 MHz, CDCl₃)

3) Radical-mediated C(sp³)-H borylation of *n*-pentane

Using benzylidenemalononitrile (**64**) and phenyl acrylate (**66**) as typical radical acceptors instead of B₂cat₂ (**2**) under our photoelectrochemical conditions, the preferential formation of 3° C-H alkylated products **65** and **67** has already revealed that site selectivity of C(sp³)-H functionalization reaction is radical acceptor depended. In order to answer the reviewer's question about whether regioselectivity of C-H borylation is also determined by the HAT species, we further investigate C(sp³)-H borylation reaction of *n*-pentane. Considering the differential features of Cl radical (secondary C-H selectivity) and Cl-boron complex (primary C-H selectivity) in the HAT process, we compare the regioselectivity results of currently existing protocols for C(sp³)-H borylation of *n*-pentane. From the scheme below we can see that all the reactions show good distal methyl selectivity, which is mainly caused by the radical acceptor depended nature. If we look more closely, the C(sp³)-H borylation reactions show better α-C(sp³)-H selectivity in the presence of metal species. It may be due to the fact that the presence of metal species plays a key role in the cleavage of boron-boron bond of B₂cat₂ and promotes the formation of key Cl-boron complex. Considering that HAT is a fast process in our photoelectrochemical system (KIE: $k_H/k_D = 1.0$) as well as the fact that 33% yield of secondary boronate is generated, this result suggests that both free Cl radical and Cl-boron complex as HAT species could exist in our reaction system.

In addition, we found that the unconventional selectivity of C(sp³)-H borylation reaction is not only determined by HAT species or radical acceptor. There are many other factors that could greatly affect the site selectivity of the photoelectrochemical reaction, such as equivalents of borylating

agent, alkane substrate, HCl, and concentration of the reaction. Please see the figure below, as well as more details in the revised SI (2.12. *Regioselectivity studies with n-pentane*).

4) Conclusion

Based on the above results, we propose that free Cl radical-mediated and Cl-boron complex-mediated HAT processes are both likely to exist in our photoelectrochemical system. We have corrected the proposed mechanism, as shown in the scheme below as well as in the revised manuscript (see Fig. 6). On the basis of fast and reversible HAT and the trapping experiments with benzylidenemalononitrile and phenyl acrylate, we believe that the free Cl radical-mediated HAT is the dominant during our photoelectrochemical reaction process.

Based on the above mechanistic studies and relevant literature reports, we propose the plausible mechanism outlined in Fig. 6. First, the Cl^- is electrochemically oxidized to Cl_2 at the anode surface, and then the chlorine radical species (**A**) is generated via light-promoted homolytic cleavage of Cl_2 . According to its inherent selectivity, the chlorine radical undergoes HAT process with $\text{C}(\text{sp}^3)\text{-H}$ compounds (taking **S9** as an example) to initially release a more substituted carbon-centered radical (**B**), which could not proceed with the C-H borylation reaction at the tertiary site, probably due to steric hindrance. Since the alkyl radical formation is reversible (or the alkyl radical can "isomerize" by performing the subsequent reversible HAT with other molecules of the substrate) and fast, a sterically unhindered primary radical (**D**) is generated and trapped by $\text{B}_2(\text{cat})_2$ to give an alkyl boronate ester (**E**) as well as the ligated boryl radical (**F**) (*Path A*). Alternatively, the HO-Bcat (**G**) generated by hydrolysis and anodic oxidation in the reaction system can be complexed with Cl radical species (**A**) to obtain Cl-radical-boron "ate" complex (**H**), followed by the HAT process to obtain carbon center radical (**D**) (*Path B*). Treating intermediate **E** with pinacol and triethylamine finally delivers the desired product **9**. At the cathode surface, protons undergo cathodic reduction to generate H_2 , obviating the need for sacrificial oxidants.

2. From a methodology aspect, this research involves a known photoelectrochemical strategy (Ref: 14) for radicals generation. Moreover, the unconventional selectivity for radical borylation has been reported in several reports (e.g. ref 64, 65). It is not significant to the field and related fields, unless the above-mentioned questions in 1 could be answered. Otherwise, this investigation is just an synthetic extension of ref 14.

Response: We are very grateful for the reviewer's comments. As far as we know, the direct one step photoelectrochemical transformation of simple hydrocarbons into alkyl boronates was previously not known, and our protocol shows the advantages of simple conditions and scalability in comparison to the current $\text{C}(\text{sp}^3)\text{-H}$ borylation protocols using stoichiometric oxidants (*Nature* **2020**, 586, 714–719; *J. Am. Chem. Soc.* **2023**, 145, 7600–7611) or stoichiometric metal catalysts (*Chem. Commun.* **2023**, 59, 7108–7111; *J. Am. Chem. Soc.* **2023**, 145, 15207–15217). Furthermore, in-depth mechanistic studies including the exploration of the HAT species and site selectivity for $\text{C}(\text{sp}^3)\text{-H}$ borylation have been extensively carried out. Given the significance and versatility of organoboron compounds in organic synthesis as well as the novel feature of photoelectrochemical transformation, we really appreciate it if you could reconsider our research work for publication in *Nature Communications*.

Reviewer #2 (Remarks to the Author):

The authors have addressed previous comments. To ensure replicability of the outcomes, it is crucial to furnish comprehensive details about the flow reactor, including its design, operating conditions, and specific parameters. This will facilitate an accurate recreation of the experiment in different settings.

Response: We are very grateful for the reviewer's positive comments and great suggestion. We have provided the details about the flow reactor, including its design, operating conditions, and specific parameters. All these information have been added into the revised SI.

Procedure for gram-scale reaction in continuous flow

Note: The design is shown in Figure S3. The flow electrolysis cell is assembled using two aluminum bodies (① and ⑤, 120 mm x 110 mm x 8 mm) with a groove (70 mm x 60 mm x 8 mm). The main material of ② is translucent quartz glass (70 mm x 60 mm x 3 mm). ③ is a reaction module with internal reaction electrodes (⑥ With grooves, 70 mm x 60 mm x 5 mm), and the external material is mainly polytetrafluoroethylene. ④ is a Graphene module with grooves (Thickness is 10 mm). flow reactor: 120 * 110 * 50mm.

Figure S3. Flow reactor

Figure S4. Flow reaction device setup

To ensure that the system is in an oxygen free state, a pump is first used to fill the entire pipeline with acetonitrile in a nitrogen atmosphere, during which a high flow rate is required to eliminate bubbles. Then place the degassed reaction bottle into the reactor. Electrolysis is a Electrolytic cell equipped with graphite anode and graphite cathode. The constant current is 10 mA, the volume is 6 cm³, and the distance between electrodes is 2mm. Use a pump to flow through the Electrolytic cell at a flow rate of 0.1 mL/min. Use a dry Round-bottom flask at the outlet to collect the reaction liquid, and then use Pinacol and triethylamine for post-treatment.

$I = 10 \text{ mA}$, 390 nm LEDs (50 W), flow rate: 0.1 mL/min

By applying a flow rate of 0.1 mL min⁻¹, an oven dried 250 mL round bottomed flask was charged with a stir bar. The reaction mixture consisted **2** (7.76 g, 20.0 mmol), Et₄NCl (1.68 g, 10 mmol), HCl (concentrated, 5.0 mL, 60 mmol), **1** (21.6 mL, 200 mmol), MeCN (150 mL). The reaction mixture was stirred at 900 rpm, irradiated with 390 nm LED (50 W) lamps and 10.0 mA electrolysis. Use a fan to blow to approximately room temperature. After irradiation, a solution of pinacol (7.08 g, 60 mmol, 3.0 equiv.) and Et₃N (40.4 mL, 400 mmol, 20 equiv.) in CH₂Cl₂ (20 mL) was added and stirring was continued for 1 h. The reaction mixture was concentrated in vacuo and then purified by flash column chromatography (10% EtOAc/henane) to give **3** (2.56 g, 12.2 mmol, 61%) as a colourless oil.

By applying a flow rate of 0.1 mL min^{-1} , an oven dried 250 mL round bottomed flask was charged with a stir bar. The reaction mixture consisted **2** (7.76 g, 20.0 mmol), Et_4NCl (1.68 g, 10 mmol), HCl (in 1,4-Dioxane, 15.0 mL, 60 mmol), **S38** (28 mL, 200 mmol), MeCN (150 mL). The reaction mixture was stirred at 900 rpm, irradiated with 390 nm LED (50 W) lamps and 10.0 mA electrolysis. Use a fan to blow to approximately room temperature. After irradiation, a solution of pinacol (7.08 g, 60 mmol, 3.0 equiv.) and Et_3N (40.4 mL, 400 mmol, 20 equiv.) in CH_2Cl_2 (20 mL) was added and stirring was continued for 1 h. The reaction mixture was concentrated in vacuo and then purified by flash column chromatography (10% EtOAc /hexane) to give **38** (2.69 g, 11.2 mmol, 56%) as a colourless oil.

Figure S5. Design of the flow reactor for the gram-scale reaction

REVIEWERS' COMMENTS

Reviewer #1 (Remarks to the Author):

In this revision, Professor Xia and the coauthors provided a series of thorough and convincing mechanistic studies to support the revised mechanism. Especially, the mechanistic studies clearly indicated the possibility of both Cl and the complex as HAT reagents, which was the major concern from the last review.

Herein, the reviewer supports the acceptance of this manuscript in Nature Communications.

Reviewer #2 (Remarks to the Author):

The authors have addressed the comments. This work is recommended for publication.

REVIEWERS' COMMENTS

Reviewer #1 (Remarks to the Author):

In this revision, Professor Xia and the coauthors provided a series of thorough and convincing mechanistic studies to support the revised mechanism. Especially, the mechanistic studies clearly indicated the possibility of both Cl and the complex as HAT reagents, which was the major concern from the last review.

Herein, the reviewer supports the acceptaion of this manuscript in Nature Communications.

Response: We are very grateful to the reviewers for agreeing to publish our manuscript.

Reviewer #2 (Remarks to the Author):

The authors have addressed the comments. This work is recommended for publication.

Response: We are very grateful to the reviewers for agreeing to publish our manuscript.